# Can Essential Oils Be a Natural Alternative for the Control of *Spodoptera frugiperda*? A Review of Toxicity Methods and Their Modes of Action

**DOI:** 10.3390/plants12010003

**Published:** 2022-12-20

**Authors:** Virginia L. Usseglio, José S. Dambolena, María P. Zunino

**Affiliations:** 1Instituto Multidisciplinario de Biología Vegetal (IMBiV-CONICET-UNC), Córdoba X5016GCN, Argentina; 2Cátedra de Química General, Faculta de Ciencias Exactas, Físicas y Naturales (FCEFyN-UNC), Córdoba X5016GCN, Argentina; 3Instituto de Ciencia y Tecnología de los Alimentos (ICTA-FCEFyN-UNC), Córdoba X5016GCN, Argentina; 4Cátedras de Química Orgánica y Productos Naturales (FCEFyN-UNC), Córdoba X5016GCN, Argentina

**Keywords:** natural insecticides, insect pest, phytochemicals, mortality rate

## Abstract

*Spodoptera frugiperda* is a major pest of maize crops. The application of synthetic insecticides and the use of Bt maize varieties are the principal strategies used for its control. However, due to the development of pesticide resistance and the negative impact of insecticides on the environment, natural alternatives are constantly being searched for. Accordingly, the objective of this review was to evaluate the use of essential oils (EOs) as natural alternatives for controlling *S. frugiperda*. This review article covers the composition of EOs, methods used for the evaluation of EO toxicity, EO effects, and their mode of action. Although the EOs of *Ocimum basilicum*, *Piper marginatum*, and *Lippia alba* are the most frequently used, *Ageratum conyzoides*, *P. septuplinervium*. *O. gratissimum* and *Siparuna guianensis* were shown to be the most effective. As the principal components of these EOs vary, then their mode of action on the pest could be different. The results of our analysis allowed us to evaluate and compare the potential of certain EOs for the control of this insect. In order to obtain comparable results when evaluating the toxicity of EOs on *S. frugiperda*, it is important that methodological issues are taken into account.

## 1. Introduction

The demand for food commodities has increased exponentially with population growth. It is estimated that the world population will be between 9.4 and 10.1 billion people in 2050 [1], implying a 35% increase in the demand for food [2]. Maize (*Zea mays* L.) is among the most cultivated cereals in the world, with a global production of 1185.90 million metric tons being expected in 2022–2023 [3]. However, crop losses occur due to the action of pests, such as insects and fungi [4]. To maximize crop yields, pest control currently involves the application of approximately 2 million tons of synthetic pesticides per year, of which 29.5% corresponds to insecticides [5].

The fall armyworm (FAW) *Spodoptera frugiperda* (J. E. Smith) (Lepidoptera: Noctuidae) is among the most common pests of maize plants in the tropical regions of the Americas [6]. In addition, as a result of the expansion of agricultural frontiers, it is now considered to be an invasive pest in African countries, China, India and Australia [7,8,9,10,11,12]. *Spodoptera frugiperda* is a holometabolous insect [6,13]. From its eggs, the first of six larval stages emerges. These larvae are initially light green, after which they become dark green with three longitudinal yellowish and dark brown lines. Then, 15–25 days after emergence from the egg, the sixth stage larvae pupate, preferably in the soil for between 7 and 13 days, until completing the cycle again with the emergence of new adults [6,13].

The use of Bt maize varieties designed to resist chewing phytophagous insects, such as FAW, has been carried out since 1996 [14]. However, due to the continuous use of these maize varieties, numerous instances of resistance of *S. frugiperda* to the Cry1 protein of *Bacillus thuringiensis* (Bt) have been reported [15,16,17,18,19,20,21,22,23]. Therefore, the search for new pest control strategies has to be continued. Related to this, the search for sustainable alternatives is becoming more ever popular [15], with the exploitation of natural products obtained from the secondary metabolism of plants being an ecofriendly attractive alternative [20,21,22].

Essential oils (EOs) are a possible source of novel pesticides, due to the fact that they have contact, fumigant, attractant and repellent activities against several insect pests [24,25,26,27,28]. For example, the EO extracted from the aerial parts of *Seriphidium brevifolium* was reported to be toxic for *Aedes albopictus*, a vector of several human and domestic animal diseases [29], with *Pimpinella anisum* EOs causing toxic effects on the currant-lettuce aphid *Nasonovia ribisnigri* [30]. In addition, the *Rosmarinus officinalis* EO showed a high fumigant toxicity (Lethal concentration 95 (LC_95_) = 54.30 μL/L) and repellent effects (RI_0.__2μL/L_ = −53.42 ± 12.54 μL/L) against the weevil *Sitophilus zeamais* [27]. Specifically, these insecticidal and repellent effects against pest insects reported for different EOs could indicate their potential value in insect control. Although there are numerous review articles that have addressed the effect of EOs on several pests [31,32,33,34,35,36,37], there are no reviews that have also analyzed the toxicity of all the EOs tested on FAW. Moreover, numerous articles have principally studied the mortality caused by the EOs, and their main pure components, on other pest species of the genus *Spodptera*, *S. exigua*, *S. littoralis* and *S. litura* (Appendix A). Thus, the aim of this review is to examine the use made of EOs and their main components as natural alternatives for the control of *S. frugiperda*. Although many articles have addressed the study of sublethal effects of EOs on the fecundity, development and feeding of this pest, in this present review, the focus is placed on the determination of the insecticidal effects of EOs. To carry this out, a survey was performed of relevant published research articles. In total, 2362 research articles were evaluated, but only 27 of these contained information concerning the effects of EOs against *S. frugiperda*. The selection criteria for articles to be included in our analysis were: (1) research articles had to be published in scientific journals; (2) EOs had to be obtained by cold pressing or hydrodistillation—solvent extracts were not considered; (3) only mortality data were considered, with the effects on reproduction, development or feeding parameters not being included in our analysis; (4) toxicity data on cell cultures were not considered; (5) essential oils evaluated as nanoformulations or nanoencapsulates were not included in our analysis.

All the relevant information found is reported in the Appendix A. Our analysis of the methods used to assess toxicity, expression of results, positive controls, and mechanisms of action yielded very different results, making it difficult to make comparisons. Consequently, in this work, we propose the standardization of what we consider to be the best and most widely used method for assessing toxicity in FAW, which could then be utilized in the design and development of future studies on this pest insect.

## 2. Essential Oils Evaluated against *Spodoptera frugiperda*

An analysis was carried out which indicated the 11 most used plant families for obtaining EOs for toxicological studies on FAW (Table 1). More than 50% of the 27 selected articles referred to EOs obtained from just three plant families: Piperaceae, Lamiaceae and Verbenaceae, in order of decreasing frequency.

In total, 21 plant genera were identified in the bibliographical analysis (Table 2). The most frequently studied genus was *Piper* (13), from the Piperaceae family, followed by the genera *Ocimum* (9) and *Lippia* (9), both from the Lamiaceae family. In all, the evaluation identified a total of 57 plant species (Table 3). Of these, *Ocimum basilicum*, commonly known as “basil”, was the most used species, followed by *Piper marginatum*, the “marigold pepper, Ti Bombé or Hinojo”, and the “purple sage” *Lippia alba*.

Most of the EOs used were extracted from the aerial parts of the plants, mainly the leaves, which were used dry or fresh. In general, the EOs were obtained from the aerial plant parts by the steam dragging distillation technique. However, EOs were obtained by cold pressing when extracted from the shell of the fruits belonging to the *Citrus* species. It is interesting to note that 35% of the articles analyzed (27) did not specify the plant organ used for the EO extraction.

From the compositional analysis of the 57 EOs tested, 56 different main organic compounds were identified. The molecular structures of compounds mentioned in more than 5% of the literature are shown in Figure 1, with geranial, geraniol, linalool, α-pinene and limonene being the most frequent. The volatile organic compounds (VOCs) cited as being the major components of the 57 EOs used as insecticides against FAW are listed in Appendix A. However, only 31% of the 27 articles analyzed complemented their toxicity studies with the use of pure EO compounds. Thymol and linalool were the most evaluated EO compounds (used in 17% of the studies), followed by limonene and geraniol (12%) (Figure 1). Two of the most widely used VOCs, linalool and limonene, corresponded to the main components of EOs (Appendix A).

Commercial insecticides can be a very useful tool for comparing the insecticidal effect of EOs. Nevertheless, only 41.6% of the articles used a commercial synthetic insecticide as the positive control, with 4% using the commercial natural insecticide neem extract (*Azadirachta indica*). Deltamethrin (12.5%), a synthetic pyrethroid pesticide used in livestock, aquaculture and agriculture due to its low residue and high toxicity, as well as because of its great efficacy, was the most frequently used synthetic insecticide as the positive control [38] (Table 4). Concerning the negative control, the one most used was acetone.

In the remaining 54.2% of the articles, positive controls were not used.

## 3. Routes of Entry of Essential Oils

The physicochemical properties of the EO molecules modulate the routes of entry into the organism [39,40]. EOs are lipophilic complex mixtures of hydrocarbon compounds of 10 to 15 carbon atoms with different functional groups, such as phenols, aldehydes, ketones, alcohols and hydrocarbons [24]. Lipophilicity is among the most important parameters to take into account when selecting bioactive compounds and the methods to test them, because the insect cuticle forms a physical defense barrier [41,42,43]. Thus, the lipophilicity property of EOs makes it easier for them to reach their target within the body [43,44,45,46]. It is widely known that organophosphate insecticides, such as Dichlorvos (DDVP), penetrate through the integument until they reach the hemolymph and, subsequently, their site of action [47,48]. In turn, there is a correlation between resistance to insecticides and cuticular penetration [49,50,51]. The non-polar nature of the insect cuticle, composed mainly of aliphatic hydrocarbons, chitin and waxes, could favor the entry of lipophilic compounds, such as those present in EOs [35,42,49,52]. Thus, this is a critical property to be considered when choosing a method to assess the toxicity of EOs on *S. frugiperda.*

Another important factor to consider in EOs is their high volatility. Therefore, the way of applying the EOs and their persistence over time must be considered when evaluating their toxicity, not only in terms of the method of application, but also of the development temperature of the test [53]. Related to this, Papachristos and Stamopoulos [54] were the first to determine the importance of the temperature at which the test is carried out on the rate of vapor release and the absorption levels of EOs, and also on the effectiveness of the enzymatic machinery for detoxification of insects.

By considering the physicochemical properties of EOs, three main routes of access of these to the target insect could be determined (Figure 2A)—ingestion, inhalation and direct contact with the integument.

## 4. Toxicological Methods against *Spodoptera frugiperda*

The methods used to study the insecticidal effects of EOs on FAW are shown in Figure 3. Thirty-three toxicity assays were identified in the analysis of the articles. Of these assays, the most cited method for studying toxicity was the topical application of insects (18), followed by toxicity by ingestion (8) and contact toxicity (4) techniques, with fumigant toxicity (2) and immersion (1) being the least used methods. It should be noted that the methodologies usually differed slightly between articles.

Usually, the topical application technique consists of applying the EOs (1 μL solutions) topically on the second thoracic segment of the larvae. Then, these larvae are separated and placed with food in different containers, such as Petri dishes, flasks or microplates, to avoid cannibalism. The incubation temperature and humidity conditions normally used for the test are 25.5 ± 1.6 °C and 68.7 ± 7.3% RH, respectively, with the temperature and humidity ranges reported being 25–28 °C and 65–70% RH, respectively, and with a photoperiod of 12 h/12 h. The evaluation of mortality is normally carried out 48 h after the start of the experiment, and the results are expressed, in most cases, as a function of the lethal dose (LD) 50 or 90. When the lethal dose could not be calculated, the articles expressed the mortality was reported as a percentage.

The toxicity of EOs was mainly tested on larval stages (96%), while a few articles (4%) evaluated the toxicity on eggs. The FAW has six different larval stages (Figure 2B). Of these, 61% of the EO toxicity studies were carried out on the third stage, while the remaining ones were performed on the second (22%), first (14%) or fourth (3%) stages. There were no studies reporting EO toxicity being carried out on the fifth or sixth stage larvae. This is in agreement with numerous manuals about the control of *S. frugiperda*, which have indicated that 4 to 10 days after oviposition is the optimal time to apply chemical controls (Figure 2B) because the larvae are newly hatched, and also to minimize the damage that these insects can cause to the maize crop [55,56,57,58].

It is interesting to note that only 9% of the studies evaluated the toxic effects of EOs in the absence of food, with 78% of the studies being carried out in the presence of artificial diet and 4% using maize or rice leaves as a food source. In all cases, the artificial diet was made up from the following ingredients: beans; wheat germ; brewer’s yeast; sorbic acid; ascorbic acid; methylparaben; agar; formaldehyde; and preservative solution (composed of propionic acid, phosphoric acid and water).

## 5. Toxicity of Essential Oils against *Spodoptera frugiperda*

A high variability in the mortality results was observed among the articles, which constituted a drawback for comparing the effectiveness of EOs. Thus, in order to make a valid comparison of the toxic effects, we transformed the EO toxicity results, with the results of the topical application method, the most cited, being shown in Table 5. For this method, the EO of *Ocimum gratissimum* (LD_50_ = 2.5 × 10^−4^ mg/g insect) was the most toxic as a topical agent [59] (Table 5), but Cruz et al. [60] reported an LD_50_ of 1.52 mg/g insect for the same EO. On analyzing the chemical composition of both EOs, it can be seen that the variety used by Monteiro et al. [59] had thymol as its main component, while for the EO used by Cruz et al. [61], the main component was trans-anethole. This variation in the results allows us also to highlight the importance of using the correct identification of the plant species and the variety that was used to carry out this type of test. Moreover, EOs can present differences in chemical composition depending on the environment in which the plant develops, thereby generating different chemotypes for the same plant species [62,63,64,65], and these small variations can result in marked differences in their bioactivities [66,67,68]. Interestingly, Lima et al. [69] reported three different LD_50_ values for the EO of *P*. *hispidinervum*. The lethal dose calculated at 48 h after application was the most toxic (LD_50_ = 3.39 mg/g insect), followed by the dose at 96 h (LD_50_ = 3.56 mg/g insect), despite no significant differences being found for the LD_50_ determined for 48 h. The lethal dose calculated at 24 h was being the least active (LD_50_ = 4.62 mg/g insect) and was statistically different from the LD_50_ obtained for 48 and 96 h.

Another method used was the contact toxicity method, in which EOs are applied to a filter paper. For this method, only three EOs were tested, with the most toxic EO of these being *Siparuna guianensis* (LC_50_ = 0.034 μL/cm^2^ and 0.038 μL/cm^2^ for susceptible and Bt-resistant strains, respectively [79]) (Table 6).

It is useful to highlight the differences in the type of insect strain used by Lourenço et al. since the responses obtained for the same EO can vary [79]. In addition, given that among the most widely used methods for the control of this insect is the utilization of transgenic varieties of maize, it would be interesting to observe how larvae resistant to the Bt toxin would respond to alternative control methods, because genetically modified (GM) maize is the main control method used for this insect and, as mentioned above, the first cases of resistance to it have already been reported [15,16,17,18,19,20,21,22,23].

On the other hand, for the ingestion toxicity methods, the EOs of *Ageratum conyzoides* (LC_50_ = 3.43 ppm) and *Piper hispidinervum* (LC_50_ = 9.4 mg/mL) were the most toxic essential oils when applied by immersion of the maize or rice leaf in solutions of increasing EO concentrations [69,80] (Table 7). In contrast, *Cymbopogon citratus* (LC_50_ = 0.19 μL/cm^2^) was the most effective EO when applied on the ventral part of the leaf [71], with *Citrus limon* revealing the highest toxic effect when the EO was mixed with the artificial diet (98.29 ppm) [81]. As no toxicity was observed in the artificial diet mixed with *C. sinensis* EO, its LC_50_ could not be calculated [82].

The fumigant effect was determined only for four EOs (Table 8), with the highest fumigant toxicity being observed for the EO of *P. septuplinervium* (LC_50_ = 9.4 μL/L of air) [84], followed by that of *P. subtomentosum* (LC_50_ = 13.2 μL/L of air) [84]. It is interesting to note that in the same study the lethal concentrations of the *P. subtomentosum* EO were very dissimilar depending on the part of the plant from which the oil was extracted, with EOs extracted from the inflorescence of *P. subtomentosum* being 12 fold more toxic than EOs extracted from leaves of the same species. Therefore, this variability in the biological activity may be attributed to differences in the chemical composition of each EO when extracted from different plant organs [85,86,87]. For this reason, it is important that information is provided not only about the method used to test the EO toxicity, but also on the plant organ from which it is extracted, the extraction method used, and its chemical composition.

Only the *P. marginatum* EO was tested by immersion of the insects in the EO solution, and the lethal concentration obtained was 152.95 ppm [88]. Furthermore, this was the only method that was performed using *S. frugiperda* eggs.

The results obtained from the analysis of the methodologies used to test the toxicity of EOs can be used as background knowledge for the standardized design of topical application methods that facilitate the comparison of the registered effects. In turn, this background serves as a justification for the selected larval stage, exposure time, laboratory conditions, and the way in which the toxicity results are expressed, among other factors. As can be seen in the previous tables, the most used larval stage for evaluating the lethality of EOs is the third, followed by the second, and finally the first, which is in agreement with the time proposed by the Insect Resistance Management Program (MRI) and the Insecticide Resistance Action Committee (IRAC Argentina) [55] for the chemical control of this pest insect. Moreover, the preferred exposure time chosen for the studies is 48 h, when the most toxic effects are observed. Lastly, the most recommended way of expressing the mortality results obtained is as a function of the lethal dose or concentration, in order that the results can be compared. Considering that the most widely used method is that of topical application, the recommended units to express the results are mg, ug or uL of EO per g of larva or per larva, if their average weight is indicated.

### 5.1. Volatile Organic Compounds Tested against Spodoptera frugiperda

Of the 37 pure compounds tested in the articles analyzed, 20 of these compounds were of natural origin, while 17 were compounds of synthetic origin, more specifically of commercial insecticides. The topical application method was the most frequently used technique for assessing the toxicity of pure compounds (78.4%), followed by the contact method (10.8%), fumigant method (8.1%) and immersion (2.7%). It should be noted that the toxic effect of the pure compounds was not evaluated by the ingestion method.

The lethal doses obtained by the topical application method are shown in Table 9. Fourteen of the tested compounds were natural ones, while eleven were synthetic insecticides. The most effective natural compound identified was trans-anethole (LD_50_ = 0.027 mg/g of insect), followed by citronellal (LD_50_ = 0.07 mg/g of insect). Trans-anethole has been widely reported for its toxic effects on arthropod pests such as *N. ribisnigri*, the currant-lettuce aphid, *T. castaneum*, *S. oryzae*, *Hyphantria cunea*, the American white moth, and *Cryptolestes ferrugineus* [30,89,90,91,92]. Both linalool and thymol were tested by different authors using the same method, but with dissimilar results being reported. For linalool, Silva et al. [93] obtained a LD_50_ of 2.10 mg/g of insect, while Cruz et al. [61] obtained a LD_50_ 2.5-fold higher (LD_50_ = 5.20 mg/g of insect). For thymol, the lethal doses obtained by Monteiro et al. [59] and Lima et al. [74] were similar, but their confidence limits indicated statistically significant differences (LD_50_ = 3.19 mg/g of insect and 4.91 mg/g of insect, respectively) (Table 7).

It is interesting to note that the lethal doses determined for the synthetic insecticides were between 2 and 3 fold less than those reported for the natural compounds (Table 9), with γ-cyhalothrin being the most toxic synthetic insecticide (LD_50_ = 1.4 × 10^−5^ mg/g of insect). For the synthetic insecticides, the 50% lethal doses determined by different authors did not show a high variability. For example, for deltamethrin Silva et al. [93], Silva et al. [77] and Lima et al. [74] reported very similar LD_50_ values (LD_50_ = 2.45 × 10^−4^, 2.46 × 10^−4^ and 3.07 × 10^−3^ mg/g of insect, respectively).

In addition, the 50% lethal concentration (LC_50_) values calculated for each of the pure compounds for the contact, fumigant and immersion toxicity methods are shown in Table 10. Thymol was the most toxic for the contact method (LC_50_ = 0.255 μL/cm^2^) while α-pinene (LC_50_ = 0.0066 μL/L) was the most toxic for the fumigant method. Geraniol was the only pure natural compound tested by immersion (LC_50_ = 3793 ppm). A synthetic insecticide was tested in the case of the contact method, with a lethal dose of 9 × 10^−4^ μL/cm^2^ being obtained for Cry1A.105 and Cry2Ab susceptible strains of FAW, while for the resistant strain, the lethal dose obtained was almost double this value (1.5 × 10^−3^ μL/cm^2^). However, these values were not statistically different.

When the main components of the most effective EOs were analyzed for each of the toxicity methods studied, only five of these had been individually evaluated against *S. frugiperda* (Table 11). Limonene was the only pure compound tested that forms part of the composition of the EO of *C. limon*, among the most toxic EOs by ingestion. However, in its pure form, when tested by topical application (Table 9), it demonstrated a low toxic effect (LD_50_ = 32.24 mg/g of insect) [61]. In contrast, the main components of the most toxic EO by the fumigant method (*P. septuplinervium*), the isomers α and β pinene (Table 6), presented high toxicity values (LD_50α-pinene_ = 0.0066 μL/L; LD_50βpinene_ = 0.016 μL/L) [84] (Table 10), allowing the toxic fumigant effect of the EO to be associated with the presence of these two major compounds. This EO has citronellal as the third main component, which has also shown high toxicity when used in its pure form in the topical application method (LD_50_ = 0.07 mg/g of insect) [61]. Finally, pure thymol, when tested by the topical application method, presented a medium toxicity (LD_50_ = 3.19 and 4.91 mg/g of insect) [59,74], suggesting that the effect observed with the EO of *O. gratissium* may be due, in part, to the presence of thymol. These results demonstrate the importance of knowing the chemical composition of the EOs in order to carry out subsequent experiments in which the effect of their pure components can be studied and the mode of action elucidated.

### 5.2. Comparison of Insecticidal Effects of EOs between S. frugiperda and the S. littoralis-S. litura-S. exigua Complex

When carrying out the bibliographic search on the insecticidal effects of EOs, and their main components, against *S. frugiperda*, it was observed that, in addition to the great variability in the methods implemented and the results obtained, this species had fewer toxicity studies carried out than the other pest species of the same genus. In contrast, *Spodoptera littoralis*, a generalist pest of legume crops such as *Vigna radiata* [94], is the species most frequently used for the study of the toxicity of EOs and their main components, followed by *S. litura*, the tobacco cutworm that attacks cotton, beans, tobacco, vegetables, and rice [95], and *S. exigua*, the beet armyworm [96]. Based on these observations, an analysis of the studies on toxicity in this complex of species was carried out in order to make a comparison with results observed for *S. frugiperda*. The findings of this analysis are shown in Appendix A.

For the complex integrated by the other three species of *Spodoptera*, the most popular method to evaluate toxicity of the EOs was topical application, as for *S. frugiperda*, followed by the fumigant method (Appendix A). However, in contrast to *S. frugiperda*, the mortality of these larvae exposed to EOs was mainly determined at 24 h (Appendix A), rather than at 48 h. Coinciding with that observed for *S. frugiperda*, the most used larval stage in these other species is the third (Appendix A). In turn, the families of plants most used to evaluate the toxicity of their EOs on *S. littoralis-S. litura-S. exigua* were Lamiacea, Apiacea and zingiberacea (Appendix A), while for *S. frugiperda,* these were Piperacea, Lamiaceae and Verbenacea (Table 1). Nevertheless, the most used species was *F. vulgare* belonging to the Apiaceae family for the complex composed of *S. littoralis*, *S. litura* and *S. exigua* (Appendix A), while for *S. frugiperda*, the toxicity of *O. basilicum* (Lamiaceae) was the most evaluated (Table 3).

The lethality of EOs on third-instar larvae of *S. littoralis, S. litura* and *S. exigua* evaluated by the topical application method is summarized in Appendix A. This table was made by including only those LD_50_ values that were expressed in μg/insect determined at 24 and 48 h. For *S. littoralis*, *Nepeta cataria* (Lamiacea) was the most toxic tested EO while for *S. litura,* this was *Alpinia pyramidata* (Zingiberaceae) and *R. officinalis* (Lamiaceae) for *S. exigua* (Appendix A). However, the toxicity effect of none of these was evaluated on *S. frugiperda.* Therefore, it is interesting to propose the study of these EOs on FAW. *Ocimum gratissimum* was the most toxic EO tested by topical application on *S. frugiperda*. As for *S. littoralis* and *S. exigua*, this EO belongs to the Lamiaceae family. On analyzing the LD_50_, it was observed that *S. frugiperda* is less susceptible to the insecticidal effect of EO than *S. littoralis* or *S. litura*, with *S. exigua* being the most resistant species. However, it should be noted that this comparison was made between the LD_50_ determined at 24 h for *S. littoralis, S. litura* and *S. exigua* and the LD_50_ for *S. frugiperda* calculated at 48 h, so any difference in the toxic effects may be due to this.

For the pure compounds, the most used method was also that of topical application followed by toxicity by ingestion (Appendix A) in 3-stage larvae and at 24 h. Once again, the most used species was *S. littoralis*, followed by *S. litura* and *S. exigua*. The VOC most evaluated against these three *Spodoptera* species was carvacrol, followed by eugenol (Appendix A) while for *S. frugiperda*, these were linalool and limonene. In contrast to the results observed for *S. frugiperda*, thymol was the most toxic compound for *S. littoralis* and *S. litura* (Appendix A), with the values of LD_50_ being up to 20-fold lower than for FAW. For *S. exigua*, the most toxic VOC was α-thujone, an α,β-unsatured ketone untested in *S. frugiperda*, whose LD_50_ was slightly higher than that obtained for α-cypermethrin in FAW.

In the analysis of the effect of the EOs, and their main components, on three other species of *Spodoptera*, differences were observed for the parameters evaluated and the origin of the EOs with respect to *S. frugiperda*. This comparative analysis between the four species allowed to identify plant species or VOCs whose toxic effect should be determined in FAW. Based on this, we propose the study of the insecticidal effect of EOs such as *A. pyramidata* and *R. officinalis*, and the α,β-unsaturated ketone such as α-thujone, on *S. frugiperda*, which, based on these antecedents, could have great effectiveness.

### 5.3. Structure–Activity Relationship

The percentage of oxygenated and of non-oxygenated compounds present in the most toxic EOs is shown in Table 12. The most effective EOs for topical application, fumigation, contact toxicity and immersion were comprised of more than 50% non-oxygenated terpenes, suggesting that their toxic effects are attributed to this diverse group of molecules. Although numerous articles have determined that EOs rich in oxygenated compounds are more effective than those with a higher percentage of non-oxygenated compounds, the opposite was observed in this review when analyzing the composition of the most toxic EOs. In fact, the only EO that fulfilled this premise was the *C. citratus* EO, since 79% of its composition corresponded to oxygenated compounds [71].

Information on the structure of the compounds that make up the most toxic essential oils serves as a basis for rational studies that can explain and predict bioactivities. Structure–activity relationship (SAR) studies of complex mixtures of EOs are currently being carried out in order to elucidate the parameters that provide them with their insecticidal properties (SAR), and to predict the toxicity of novel compounds (quantitative structure–activity relationship, QSAR). Although there are numerous studies that have identified natural bioactive compounds in various insects, there are few antecedents of this type for *S. frugiperda*.

Several SAR and QSAR investigations have been carried out on other insect pests. For *S. zeamais,* these studies have revealed that the toxicity of the natural compounds are related to descriptors such as LogP (the octanol–water partition coefficient) and the acidity (pKa), with these playing a key role in reaching the target site of action [97]. Moreover, in addition to these parameters, it was observed that the presence of carbonyl groups is also related to a greater insecticidal effect [98]. The presence of the hydroxyl group (OH-), carbonyl carbons, polar surface (amount of molecular surface arising from polar atoms) and aromatic ring substituents are strongly related to the insecticidal capacity of the molecules [99,100]. In addition, recent in silico molecular docking studies have revealed that the site of action of acetylcholinesterase (AChE), among the main target enzymes for the development of insecticides, has a lipophilic region at its site of action [101,102], in which non-oxygenated terpenes could be interacting and may be involved in the toxic effect of these compounds. It should be noted that as these are complex mixtures of natural compounds, the effect of each of its components separately may not correlate with the effect observed as a whole.

The insecticidal effects of the α-β unsaturated ketones carvone, α and β-thujone, pulegone and thymoquinone have been widely reported on numerous pests such as *S. exigua*, *S. litura*, *S. zeamais*, *Solenopsis invicta*, and *Trioza erytreae* [103,104,105,106,107,108,109,110]. It is interesting to note that QSAR studies have determined that the efficacy of these ketone compounds is related to the presence of α-β unsaturated carbonyl carbons (ester function and a double bond conjugated to the carbonyl group) in its structure [105]. This structural configuration determines the interaction of the carbonyl group, through the Van der Waals force, with the active site of AChE, indicating that these compounds present a great insecticidal effect [111]. These antecedents could imply that unsaturated α-β ketone compounds have a similar toxic effect on *S. frugiperda,* since they act on a highly conserved site of action between insect groups. However, there are few articles that have addressed the study of this class of molecules. In fact, only one article has studied the toxic effect of carvone on FAW [73]; thus, we suggest that rational studies focused on the insecticidal evaluation of α-β unsaturated ketones should now be carried out to corroborate the effectiveness of these compounds on this insect pest. Moreover, in the rational search for bioactive molecules, future studies on structure–activity should be performed about the toxicity of natural compounds against *S. frugiperda*.

## 6. Mode of Action of Essential Oil in Insects

Lucia and Guzman [112] proposed that the mode of action of VOCs on insects is varied, with the nervous system being the main target. Neurotoxic effects can be observed in the evolution of GABA receptors, the modulation of the synapse by octopamine, and the inhibition of the enzyme acetylcholinesterase.

### 6.1. Acetylcholinesterase (AChE)

Acetylcholinesterase (AChE) is an enzyme responsible for catalyzing the hydrolysis of the neurotransmitter acetylcholine to choline and acetate in the synaptic cleft, thereby preventing its accumulation at the nerve terminal (Figure 4) [99,100,101,113]. An accumulation of this neurotransmitter leads to overstimulation of the nervous system, hyperactivity, paralysis, and the subsequent death of the insect [100,101].

Insect AChE has a specific cysteine residue that differs from that of the mammalian enzyme [99], with achieving its inhibition being a point of interest in the search for natural or synthetic insecticides. Therefore, AChE activity is the most studied enzymatic reaction in relation to the toxicity of insecticides [40].

Due to the complex nature of EOs, their effects usually differ from those of their main pure components, making it difficult to determine their mode of action [40]. Moreover, AChE inhibition can occur through two pathways: (1) EO components can act as competitive inhibitors with respect to acetylcholine (ACh) binding to the active sites of the enzyme; or (2) EO components can bind non-specifically to other sites on the enzyme (non-competitive inhibitors) and allosterically modify them. In the first case, the activity of the enzyme remains normal, but the reaction product (choline + acetate) is not formed, while the activity of the enzyme is modified in the second case [40,112]

De Oliveira et al. [102] showed that both the EO of *C. flexuosus* and citral (its main component) inhibited FAW AChE by 450 fold more than the methomyl insecticide used as a positive control. *H. marrubioides* (inhibitory concentration 50 (IC_50_) = 0.0510 mg/mL) and *O. selloi* (IC_50_ = 0.0660 mg/mL) EOs also inhibited the FAW AChE, in agreement with the results obtained for toxicity, with *H. marrubioides* EO being the most effective (LD_50_ = 0.24 mg/g insect) [83]. In another study, Fergani et al. [115] observed an increase of 64% in AChE activity in vitro in 3rd instars of *S. littoralis* larvae exposed to the “basil” EO (*O. gratissimum*) compared to the control, despite the fact that in *S. frugiperda* it presented a high toxicity (LD_50_ = 2.5 × 10^−4^ mg/g insect) [59]. The EO of *Thymus vulgaris*, whose main component is thymol, presented an IC_50_ of 3.17% *v/v* in *S. littoralis*, thereby allowing the authors to infer that this inhibitory effect could have been due to the high concentration of thymol, considering its known toxic effects [116]. The inhibitory effects of α-pinene, trans-anethole, and thymol were also investigated on *Ephestia kuehniella,* a lepidopteran pest of stored products [117]. Of these three compounds, the most effective was thymol (IC_50_ = 0.137 μL/L), followed by anethole (IC_50_ = 0.49 μL/L) and α-pinene (IC_50_ = 0.864 μL/L) [80], suggesting that the toxic effects of these compounds observed in *S. frugiperda* could have involved an inhibition of AChE activity. In *S. frugiperda*, thymol presents a medium toxicity, which may be due to the fact that it acts as a non-competitive inhibitor, thus decreasing the enzyme’s capacity, while anethole, which presents a high toxicity, could be directly interacting with the AChE site of action and thereby causing it inhibition. On the other hand, it is important to highlight the scarcity of studies that have addressed the toxicity caused by α,β-unsaturated ketone compounds in FAW, which, as previously mentioned, have high toxic effects on other pest insects. In these compounds the orbital electronegativity of the carbonyl group is related to AChE inhibition [105].

### 6.2. GABA Receptor

γ-aminobutyric acid (GABA) is a neurotransmitter that inhibits the nervous system by binding to its receptors (GABARs) (Figure 5), and is mainly related to olfactory processing and learning, memory and behavior in insects [118,119]. The disturbance of the correct functioning of the GABA/GABARs complex leads to a continuous excitation of the nervous system, causing stress and subsequent death [40].

As the GABA/GABAR complex presents differences between different groups of animals, it is an interesting target in the search for insecticides, which can take advantage of the different chemical sensitivities between insects and mammals [121]. In fact, the GABA/GABARs complex has been proposed as being the site of action of several EOs and their principal components in different insects [70,74,110,119,122,123]. Nevertheless, there are few studies that have directly studied the effect of EOs on GABA, with these studies generally relating behavioral aspects to the effects that bioactive molecules have on this neurotransmitter. Lourenço [79] observed that the *S. guianensis* EO reduces locomotion in *S. frugiperda*, which may be related to the activation of GABARs, as seen in honey bees [119]. In *S. exigua*, the lipophilic compounds of *Salvia hispanica* EO may increase the cuticular penetration of α-thujone, its principal compound, enabling it to reach the chloride channels regulated by GABA and disturbing their correct functioning [110]. Furthermore, the effects of pure natural compounds on the GABA/GABAR complex have been reported. Thymol was found to exhibit neurotoxic action by binding GABARs, thereby preventing the closure of sodium channels [74]. Moreover, limonene (IC_50_ = 11.37 mM), α-pinene (IC_50_ = 12.70 mM) and (−)-citronellal (IC_50_ = 24.17 mM) caused a high inhibition of the mite *Tetranychus urticae* γ-aminobutyric acid-transminase (GABA-T), the enzyme responsible for degrading the GABA neutrotransmitter [124]. These antecedents allow us to suppose that the bioactivity of the EOs observed on *S. frugiperda* could be due, in part, to the effect of these compounds on GABA. Nevertheless, there are few studies on the specific mode of action and the expression of genes associated with this complex [125,126,127]. Both essential oils and synthetic insecticides act as non-competitive allosteric modulators of GABARs by binding to them and maintaining a constant flow of chloride ions into the cell, thus keeping the nervous system in permanent excitement [40,101,113,122,128,129].

### 6.3. Octopamine

The neurotransmitter octopamine (OA) in insects is capable of influencing the neural mechanisms that generate the motor programs for different types of behavior, for example, foraging [130]. Specifically, octopamine is related to high-energy-cost activities such as oviposition, movement, appetitive stimulation, and stress response [40,101,113,131,132,133,134,135,136,137].

Octopamine possesses G protein-coupled membrane receptors (OAr) throughout the insect nervous system. OA binds to OAr and activates an adenylyl cyclase that converts ATP to cAMP, which behaves as a signal molecule that activates diverse cellular processes [40,113] (Figure 6). Although these OArs are structurally and functionally related to the α2 adrenergic receptors of mammals, they are not found in this class of animals [138], thereby making them a target of interest for the development of insecticides [113].

Numerous studies have addressed the OA/OAr complex as being a mediator of the toxicity of EOs and their main components in several pest insects. As for GABA, there are few studies that have been carried out directly on the octopamine neurotransmitter. In general, behavioral issues have been investigated with respect to the effect of these on the site of action. Lourenço et al. [79] associated a decrease in locomotion in the population of *S. frugiperda*, when exposed to the EO of *S. guianensis*, with alterations related to the neurotransmitter octopamine, among others. Moreover, the EO of *S. hispanica*, which has 1,8-cineole among its main components, acts as an antagonist of octopamine receptors of *S. exigua* [110]. Further, eugenol, among the principal components of *O. gratissimum* EO, acts in a mimetic way with octopamine, inducing cellular changes that could determine its insecticidal capacity against the weevil *Rhynchophorus ferrugineus* [139]. The effect of these EOs could be due to two modes of action: (1) blocking OArs [101]; or (2) acting agonistically on OA [113], with the latter being the most often mentioned. The effects of EOs and their main components are mainly related to alteration of the OA/OAr complex, due to the similarity of the changes that these natural compounds present at the level of cellular processes, for example, the increase in cellular calcium levels [113,140,141,142,143,144].

### 6.4. Other Modes of Action

Another mode of action of EOs reported was due to the alteration of cellular integrity through modifying the permeability of the ion channels of the cell membrane, thus interfering with cellular biosynthesis and operation [112], with the *S. officinalis* EO affecting the DNA integrity and inhibiting the electrons entering the respiratory chain in *A. aegypti* larvae of third and fourth instars [145]. β-asarone, the principal compound of *Acorus calamus* rhizome EO, has cytotoxic effects and induces apoptosis in the *S. frugiperda* cell line (Sf9) [146]. Moreover, the effect of α-pinene, trans- anethole and thymol on the antioxidant system of *Ephestia kuehniella* were evaluated, demonstrating that the superoxide dismutase, peroxidase, glutathione S-transferase and catalase activity were significant higher than the untreated larvae, indicating that its antioxidant system was overstimulated by the presence of terpenes [117]. On the other hand, the evaluation of lipid peroxidation products can give hints concerning the effects of EOs [112]. Malondialdehyde (MDA) and Conjugates Dienes (CD), among others, are indicators of oxidative stress whose levels, and those of their related enzymes, are generally increased in insects exposed to EOs [117,147,148,149]. For example, the lipid peroxidase enzyme activity showed a significant increase in *Agrotis ipsilon* larvae after 96 h of lemongrass (*C. citratus*) EO exposure [148]. In *M. domestica*, this EO also causes an increase in the lipid peroxidation reflected in an increase in the amount of MDA [149]. *Tagetes filifolia* also produces an increase in *T. castaneum* MDA content [147].

The knowledge of these alternative mechanisms of toxicity, in addition to classic neurological ones, could help in the design of natural formulations that can attack several different target sites.

## 7. Final Considerations

In this review, we have compiled information concerning the methods used to evaluate the toxicity of EOs on *S. frugiperda* carried out in laboratory conditions. The results of the analysis showed that there is a large number of EOs that have a great potential to be used as natural alternatives for the control of this pest under these conditions. However, as we observed that there is great variability in the methods implemented and also in the way of expressing the results, the ability to draw useful comparisons is very restricted. Based on this limitation, the objective of this article was to carry out a complex analysis of the toxicity methods used to evaluate the effects of EOs against *S. frugiperda*, from which conclusions could then be drawn and a standardization proposed of the main parameters that should be taken into account.

Although these natural products have a high potential for being used as biopesticides by attacking the same sites of action as those targeted by the synthetic insecticides, studies on a higher scale, such as pilot and field, should now be carried out to determine the effectiveness of these EOs under natural conditions. In turn, due to their high volatility and photodegradation, feasible ways of applying EOs must be found to enable them to be used in cultivars [150]. Currently, several studies are being performed on the formulation of biodegradable nanoemulsions or encapsulates that contain EOs, or their main components, with insecticidal activity that prolongs their effects over time and facilitates their application in the field, as an alternative technology for the implementation of biopesticides in agroecosystems. For example, the toxicity of an oil-in-water (O/W) nanoemulsion made with Tween 80 and Span 80 (non-ionic surfactants) and *Cedrela odorata* EOs against *S. frugiperda* has been determined [151]. Furthermore, it was observed that chitosan nanoparticles (CSNPs) loaded with *S. leucantha* EO decreased the activity of digestive enzymes in *S. litura*, *H. armígera* and *P. xylostella* [152]. Further, CSNPs loaded with citronella EO caused the interruption of the development of *S. littoralis* [153]. Nanostructured lipid carriers (NLCs), another innovative bioformulation, were made using 10% *w/v* lipid and 10% *w/v* oil (*L. angustifolia*). This bioformulation caused high mortality and the reduced progeny of *Aphis gossypii,* even when applying the nanocarrier alone [154]. Thus, as can be seen, nanotechnology allows the development of bioformulations with an optimum dosage of their bioactive components in order to improve agricultural productivity, thereby generating efficient and ecofriendly alternatives [155].

Despite EOs being classified by the US Food and Drug Administration (FDA) as Generally Recognized as Safe, we suggest that toxicity studies on other non-target organisms, animals and plants should also be performed [44]. In addition, phytotoxicity studies of EOs effects on the maize plant should accompany the mortality studies. The cost of producing EOs must also be taken into account when designing alternative strategies for pest control. Although this cost is high, there is a tendency for producers worldwide to change the cost/efficiency paradigm to one where the health of people, animals and the environment is a central issue [156,157]. Nevertheless, the number of natural pesticides in the market is still low, which may be due not only to the cost of production but also to the small number of studies carried out and with the results being applicable only in the short term. Other issues are the strict legislation hampering their incorporation into the market and the low persistence of their effects [32]. Using the knowledge of the main compounds that constitute the EOs and, in many cases, of their bioactivity, carrying out synthesizing of these compounds could lower the costs [158,159]. Furthermore, due to the high efficacy that has been demonstrated of these natural compounds, a future challenge would be for researchers and industries to work together to increase the scale of production of biopesticides and to insert them in the global market [32].

Based on the present findings, we list the following final considerations:The method most used for evaluating the toxicity of EOs on *S. frugiperda* was topical application, where the bioactive compound enters the organism through the cuticle. Considering the field application method of traditional insecticides, this method of topical application simulates what happens when the insect pest is found in the cultivar and is reached by traditional spraying. Thus, this testing method could be recommended for laboratory study using mainly third-instar larvae, in order to obtain comparable results with already published articles.The most effective EOs were *Ocimum gratissimum*, *Siparuna guianesis*, *Piper marginatum*, *Piper septuplinervium*, *Cymbopogon citratus*, *Citrus limon*, and *Ageratum conyzoides* for the methods of topical application, contact toxicity, immersion, fumigant and ingestion, respectively. In general, these essential oils presented a high percentage of non-oxygenated volatile compounds, with the exception of *C. citratus* EO, thereby allowing us to predict that against this insect, terpene hydrocarbon-type compounds would present a greater toxicity. However, mostly only pure oxygenated compounds have been tested, with anethole being the most toxic of these. We suggest that mixtures of lipophilic and hydrophilic compounds could have a greater toxic effect as the former act as vehicles for the latter to cross the insect cuticle and facilitate their arrival at the active site.EOs and their pure compounds are approximately 1000 to 100,000 fold less toxic than most insecticides. Although it is known that these synthetic insecticides produce health and environmental problems, it is important to highlight that the effect of natural compounds is significantly lower, so they may not be widely accepted by rural producers. However, alternatives could be implemented such as the formulation of synergistic mixtures between EOs, or their more bioactive components, and traditional synthetic insecticides, in order to reduce their applied concentrations.

Finally, we propose the use of the topical application method for the evaluation of the effect of EOs, and their main compounds, on third-instar larvae of *S. frugiperda*. In turn, we propose that the mortality records be carried out at 48 h. Lethal doses (50 or 90), should be expressed as a function of mg, ug or ul of EO per g of larva or per larva, if their average weight is indicated, to favor the comparison of results. Further, a positive control (commercial insecticide) should be included in the studies to be used as a reference of the effectiveness of the new EOs or VOCs evaluated. Based on the comparison of the toxic effects of EOs on other species of *Spodoptera*, we propose the study of EOs such as *Rosmarinus officinalis*, *Salvia hispanica*, *Nepeta cataria* and *Alpinia pyramidata*, and of α, β-unsaturated ketones, such as carvone and α-thujone, against *S. frugiperda*. Moreover, structure–activity relationship studies should be carried out in order to permit a rational search for compounds with insecticidal effects on this pest. The information reported in this review can serve as a guide for future studies on the effects of EOs against *S. frugiperda*.

## Figures and Tables

**Figure 1 plants-12-00003-f001:**
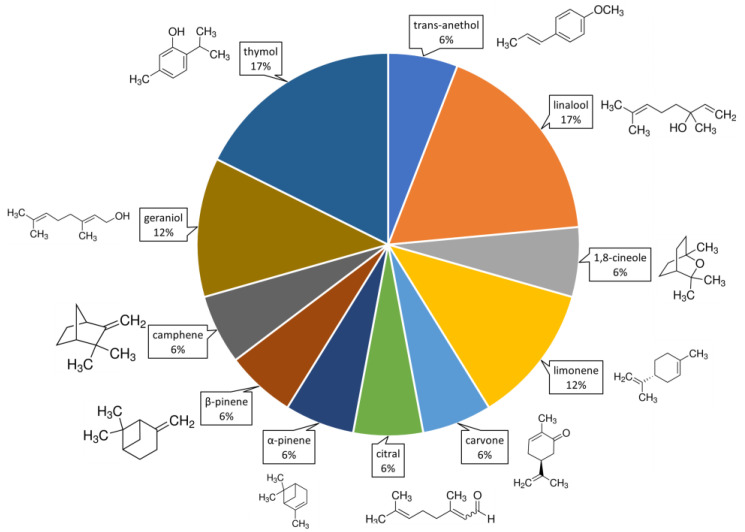
Volatile volatile organic compounds (VOCs) used to evaluate toxicity against *Spodoptera frugiperda*. The percentages indicate the proportion of appearance of these compounds in the literature. Percentages were calculated based on 23% of articles (of 27 selected) that test pure VOCs.

**Figure 2 plants-12-00003-f002:**
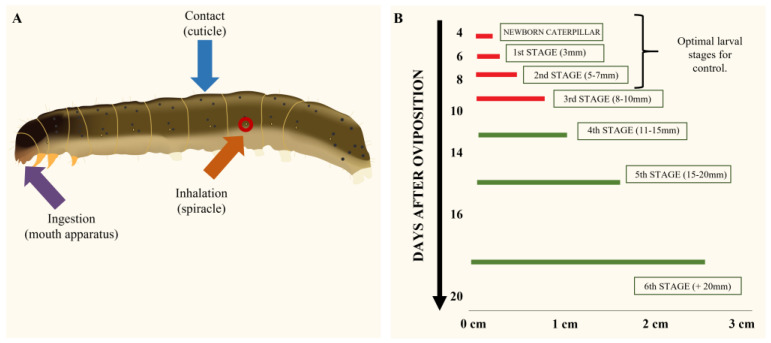
(**A**) Routes of entry of EOs to lepidopteran larvae. Orange arrow: entry through the respiratory spiracles. Purple arrow: entry through ingestion of treated food. Blue arrow: entry by direct contact with the integument. (**B**) Optimal moments of chemical control thought the larval stages of *S. frugiperda.* Red segments indicate the optimal stage for chemical control. (Modified from Programa Manejo de Resistencia de Insectos (MRI) and the Insecticide Resistance Action Committee (IRAC Argentina) [55].)

**Figure 3 plants-12-00003-f003:**
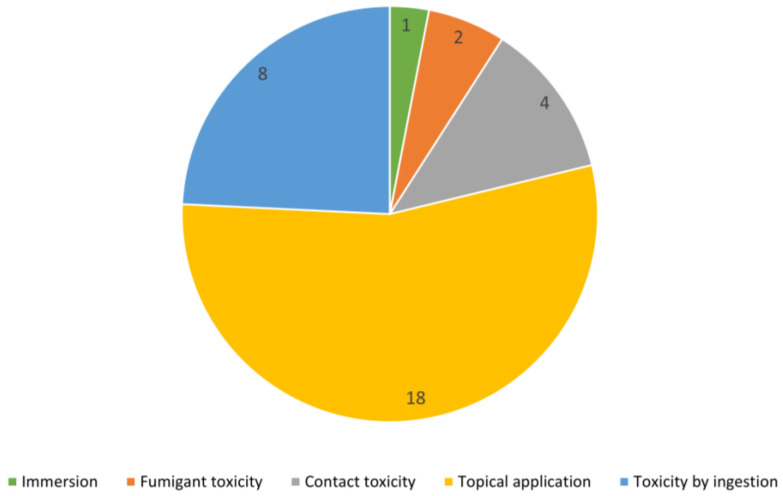
Methods used to test the insecticidal effect of EOs against *Spodoptera frugiperda*. The numbers represent the number of occurrences in the literature.

**Figure 4 plants-12-00003-f004:**
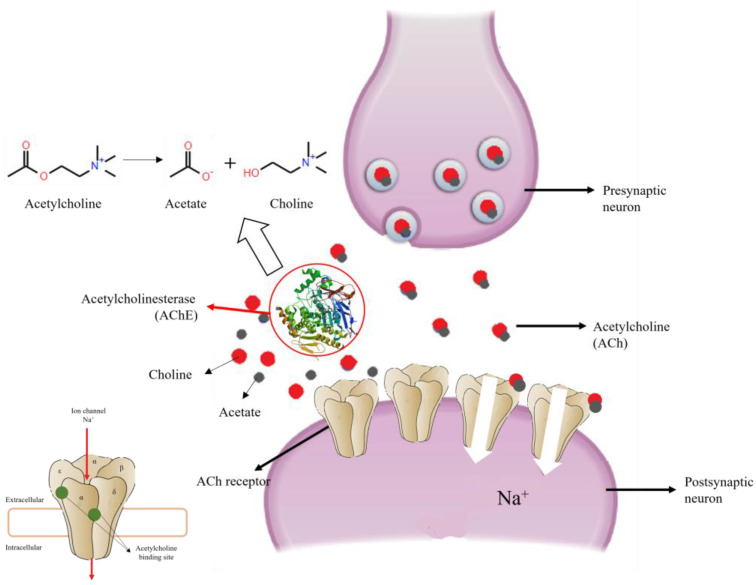
Normal functioning of the enzyme Acetylcholinesterase (AChE) in the transmission of nerve impulses. Modified from Jankowska et al. [113] and Basicmedical Key [114].

**Figure 5 plants-12-00003-f005:**
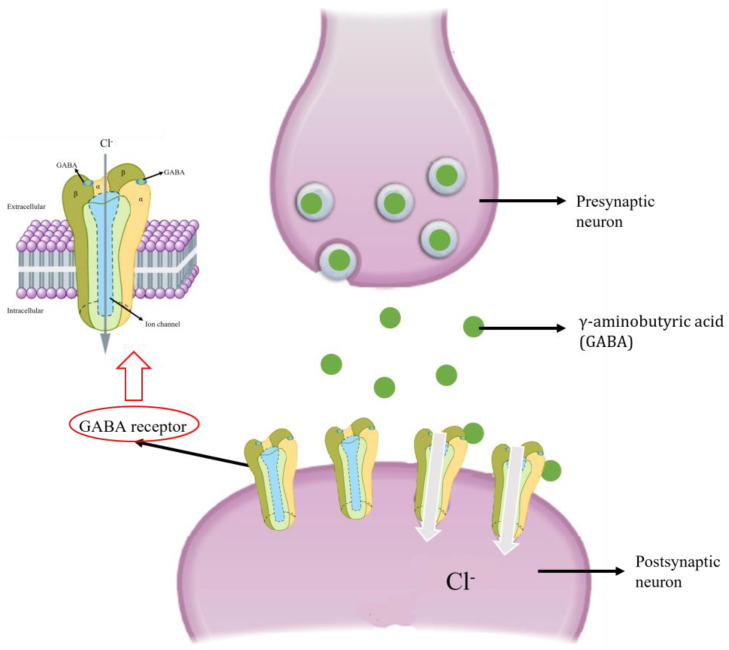
Normal functioning of the GABA/GABARs complex. Modified from Jankowska et al. [113] and Urtasum et al. [120].

**Figure 6 plants-12-00003-f006:**
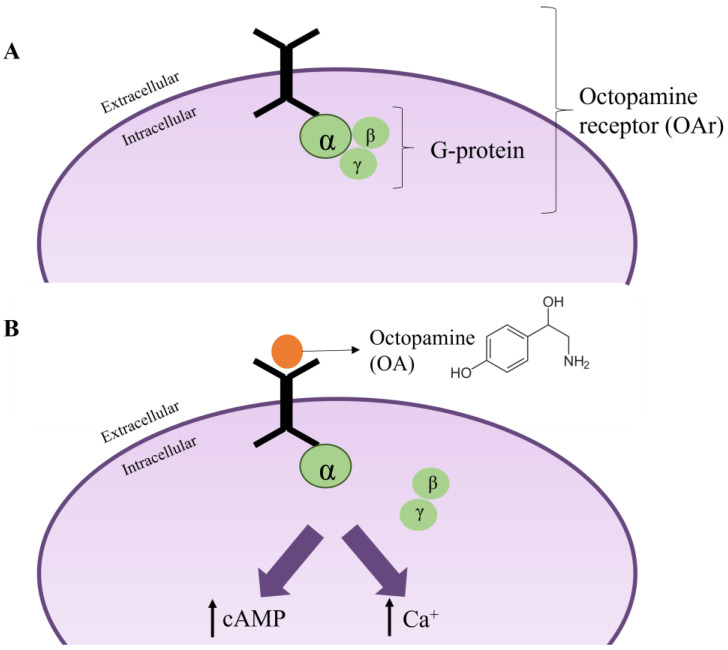
Functioning of the OA/OAr complex. (**A**) OA at rest. (**B**) Activation of OAr by OA binding. Modified from Jankowska et al. [113].

**Table 1 plants-12-00003-t001:** Occurrence of families whose essential oil has been studied as an insecticide against *Spodoptera frugiperda*.

Plant Family	Occurrence
Piperaceae	13
Lamiaceae	12
Verbenaceae	10
Myrtaceae	5
Asteraceae	5
Rutaceae	3
Poaceae	3
Zingiberaceae	2
Apiaceae	1
Siparunaceae	1
Geraniaceae	1
Total	56

**Table 2 plants-12-00003-t002:** Plant genera whose essential oils have been evaluated as insecticides against *Spodoptera frugiperda*.

Genera	Occurrence in Literature
*Piper*	13
*Ocimum*	9
*Lippia*	9
*Eucalyptus*	5
*Hyptis*	3
*Cymbopogon*	3
*Foeniculum*	2
*Corymbia*	2
*Citrus*	2
*Siparuna*	1
*Ruta*	1
*Pelargonium*	1
*Mentha*	1
*Malva*	1
*Hyptis*	1
*Eremanthus*	1
*Tanacetum*	1
*Artemisia*	1
*Ageratum*	1
*Zingiber*	1
*Vanillosmopsis*	1
Total	60

**Table 3 plants-12-00003-t003:** Plant species whose essential oils are evaluated as insecticides against *Spodoptera frugiperda*.

Plant Species	Occurrences in the Literature
*Ocimum basilicum*	4
*Lippia alba*	3
*Piper marginatum*	3
*Corymbia citriodora*	2
*Cymbopogon citratus*	2
*Eucalyptus staigeriana*	2
*Foeniculum vulgare*	2
*Hyptis marrubioides*	2
*Lippia microphylla*	2
*Lippia sidoides*	2
*Ocimum gratissimum*	2
*Piper arboreum*	2
*Piper corcovadensis*	2
*Piper hispidinervum*	2
*Ageratum conyzoides*	1
*Artemisia absinthium*	1
*Citrus aurantium*	1
*Citrus limon*	1
*Citrus sinensis*	1
*Cymbopogon winterianus*	1
*Eremanthus erythropappus*	1
*Eucalyptus citriodora*	1
*Eucalyptus urograndis*	1
*Eucalyptus urophylla*	1
*Hyptis suaveolens*	1
*Lippia gracilis*	1
*Lippia origanoides*	1
*Malva* sp.	1
*Mentha* sp.	1
*Ocimum selloi*	1
*Pelargonium graveolens*	1
*Piper aduncum*	1
*Piper septuplinervium*	1
*Piper subtomentosum*	1
*Ruta graveolens*	1
*Siparuna guianensis*	1
*Tanacetum vulgare*	1
*Vanillosmopsis arborea*	1
*Zingiber officinale*	1
Total	57

**Table 4 plants-12-00003-t004:** Commercial insecticides used as positive controls against *Spodoptera frugiperda*.

Positive Control Used in Bibliography	Occurrences in the Literature
neem extract (*Azadirachta indica*)	2
deltamethrin	3
α-cypermethrin	1
β-cypermethrin	1
fenpropathrin	1
δ-cyhalothrin	1
Indoxacarb	1
chlorpyrifos	1

**Table 5 plants-12-00003-t005:** Lethal effects of essential oils tested by the topical application method.

Essential Oil	50% Lethal Doses (LD_50_; mg/g Insect, CI)	Larval Stage	Reference
*Ocimum gratissimum*	2.5 (1.7–2.6) × 10^−4^ *^2^	3rd	[59]
*Lippia gracilis*	1.2 (0.9–1.6) × 10^−3^ *^2^	3rd	[70]
*Artemisia absinthium*	7.1 (5.3–7.2) × 10^−2^ *^2^	2nd	[71]
*Hyptis marrubioides*	0.24 (0.21–0.26) *^2^	2nd	[72]
*Ocimum basilicum*	0.49 (0.45–0.53) *^2^	2nd	[72]
*Pelargonium graveolens*	1.13 (0.083–0.145) ^2^	3rd	[73]
*Lippia alba* (LA-10)	1.2 (0.84–1.57) *^2^	3rd	[73]
*Lippia alba* (LA-57)	1.21 (0.90–1.57) *^2^	3rd	[73]
*Ocimum gratissimum*	1.52 (1.36–1.67) ^2^	3rd	[60]
*Ocimum gratissimum* (White wild basil)	1.52 (1.36–1.67) ^2^	3rd	[61]
*Lippia alba* (LA-22)	1.56 (1.18–2.02) *^2^	3rd	[73]
*Ocimum gratissimum* (Wild basil)	2.84 (2.34–3.38) ^2^	3rd	[61]
*Eucalyptus staigeriana*	3.2 (2.41–4.07) ^2^	3rd	[60]
*Lippia sidoides*	3.21 (2.95–3.49) ^2^	3rd	[74]
*Piper hispidinervum*	3.39 (3.42–4.15) *^2^	3rd	[69]
*Piper hispidinervum*	3.56 (3.22–3.91) *^3^	3rd	[69]
*Piper corcovadensis*	3.58 (nd) ^2^	3rd	[75]
*Piper marginatum*	4.18 (nd) ^2^	3rd	[75]
*Eucalyptus citriodora*	4.58 (4.09–5.08) ^2^	3rd	[61]
*Corymbia citriodora*	4.59 (4.15–5.03) ^2^	2nd	[76]
*Piper hispidinervum*	4.62 (4.10–5.22) *^1^	3rd	[69]
*Ocimum basilicum*	4.86 (4.02–6.13) ^2^	3rd	[61]
*Foeniculum vulgare*	5.05 (4.13–5.96) ^2^	3rd	[61]
*Lippia microphylla*	5.35 (4.65–6.05) ^2^	2nd	[76]
*Ocimum basilicum*	6.27 (5.80–6.73) *^2^	3rd	[77]
*Piper arboreum*	10.91 (nd) ^2^	3rd	[75]
*Piper aduncum*	12 (7.1–18.0) ^2^	3rd	[78]
*Vanillosmopsis arborea*	172.86 (152.8–200.0) ^2^	3rd	[66]

All values marked with an asterisk (*) were recalculated for comparison. ^1^ Determined at 24 h. ^2^ Determined at 48 h. ^3^ Determined at 96 h.

**Table 6 plants-12-00003-t006:** Lethal effects obtained by contact toxicity method.

Essential Oil	50% Lethal Concentration (LC_50_; μL/cm^2^, CI)	Larval Stage	Reference
*Siparuna guianensis*	0.034 (0.033–0.034) μL/cm^2^ *^1^	3rd	[79]
*Siparuna guianensis*	0.038 (0.036–0.047) μL/cm^2^ *^2^	3rd	[79]
*Ocimum gratissimum*	0.171 (0.150–0.193) μL/cm^2^	3rd	[59]
*Lippia gracilis*	1.55 (1.51–1.59) μL/cm^2^	3rd	[70]

All values marked with an asterisk (*) were recalculated for comparison. ^1^ Cry1A.105- and Cry2Ab-susceptible strain of *S. frugiperda*. ^2^ Cry1A.105- and Cry2Ab-resistant strain of *S. frugiperda*.

**Table 7 plants-12-00003-t007:** Lethal effects of the EOs present in food.

Essential Oils	EO Application Method	Lethal Concentration 50 (LC_50_)	Concentration	Mortality (%) at 96 h	Larval Stage	Reference
*Cymbopogon citratus*	On ventral part of the leaf	0.19 (0.13–0.38) μL/cm^2^			1st	[71]
*Zingiber officinale*		0.25 (0.20–0.35) μL/cm^2^			1st	[71]
*Mentha* sp.		0.33 (0.16–1.93) μL/cm^2^			1st	[71]
*Ruta graveolens*		0.62 (0.49–1.02) μL/cm^2^			1st	[71]
*Malva* sp.		0.67 (0.58–0.82) μL/cm^2^			1st	[71]
*Artemisia absinthium*		2.09 (1.64–2.96) μL/cm^2^			1st	[71]
*Citrus limon*	Mixed in artificial diet	98.29 ppm *			2nd	[81]
*Citrus aurantium*		100 ppm *			2nd	[81]
*Ocimum selloi*		600 (580–620) ppm *			3rd	[83]
*Citrus sinensis*		ND	0.1 mg/g of diet	0	2nd	[82]
			1 mg/g of diet	5	2nd	
			10 mg/g of diet	0	2nd	
			0.1 mg/g of diet	10	2nd	
			1 mg/g of diet	5	2nd	
			10 mg/g of diet	0	2nd	
*Ageratum conyzoides*	By immersion of the maize or rice leaf in EOs solutions	3430 ppm *			1st	[80]
*Piper hispidinervum*		9400 (7900–11,100) ppm			1st	[69]
		16,200 (14,400–18,400) ppm			1st	
		17000 (13,700–21,100) ppm			1st	
		17,900 (15,900–20,200) ppm			1st	
		18,200 (16,800–19,700) ppm			1st	
		28,300 (24,300–32,900) ppm			1st	

All values marked with an asterisk (*) were recalculated for comparison.

**Table 8 plants-12-00003-t008:** Lethal effects of EOs tested by fumigation.

Essential Oil	50% Lethal Concentration (LC_50_; μL/L of Air)	Larval Stage	Reference
*Piper septuplinervium* ^1^	9.4 (7.72–11.4) *	2nd	[84]
*Piper subtomentosum* ^2^	13.2 (10.3–16.6) *	2nd	[84]
*Corymbia citriodora* ^3^	44.85 (36.89–52.81) *	2nd	[76]
*Lippia microphylla* ^3^	116.52 (95.77–137.27) *	2nd	[76]
*Piper subtomentosum* ^3^	146 (116–180) *	2nd	[84]

All values marked with an asterisk (*) were recalculated for comparison. ^1^ Obtained from the aerial part of the plant. ^2^ Obtained from the inflorescence. ^3^ Obtained from the leaves.

**Table 9 plants-12-00003-t009:** Lethal effects of pure compounds tested, on *Spodoptera frugiperda*, by topical application.

Compound	50% Lethal Doses (LD_50_) (mg/g of Insect, CI)	Mortality (% ± SD) ^1^	Larval Stage	LogP	Reference
γ-cyhalothrin ^2^	1.4 (1.08–1.78) × 10^−5^ *		3rd	6.20	[78]
Chlorpyrifos ^2^	2.4 (0.83–4) × 10^−4^ *		2nd	4.77–3.71	[72]
Deltamethrin ^2^	2.45 (1.13–3.76) × 10^−4^ *		3rd	6.20	[93]
Deltamethrin ^2^	2.46 (1.14–3.78) × 10^−4^ *		3rd	6.20	[77]
Deltamethrin ^2^	3.07 (2.58–3.35) × 10^−3^		3rd	6.20	[74]
Decis 25 ^2^ (Deltamethrin)	3.17 (2.20–4.57) × 10^−3^ *		3rd	6.20	[93]
Commercial product ^2^	3.2 (2.2–4.6) × 10^−3^ *		3rd		[77]
trans-anethole	0.027 (0.021–0.032)		3rd	3.17	[61]
citronellal	0.07 (0.06–0.08)		3rd	3.48	[61]
Fenpropathrin ^2^	0.18 (0.17–0.23) *		3rd	5.48	[78]
α-cypermethrin ^2^	0.19 (0.12–0.28) *		3rd	6.27	[78]
β-cypermethrin ^2^	1.03 (0.016–1.37) *		3rd	6.27	[78]
linalool	2.10 (1.65–2.56) *		3rd	3.28	[93]
α-pinene	2.40 (2.06–2.67) *		3rd	4.37	[59]
thymol	3.19 (2.93–3.45) *		3rd	3.28	[59]
thymol	4.91 (4.35–5.56)		3rd	3.28	[74]
linalool	5.20 (4.21–6.27)		3rd	3.28	[61]
limonene	32.24 (27.73–36.55)		3rd	4.45	[61]
1,8-cineole		2.0 ± 2.0	3rd	2.82	[73]
limonene		4.00 ± 2.44	3rd	4.45	[73]
Azamax ^2^		14.00 ± 5.09	3rd		[73]
geraniol		30.00 ± 8.84	3rd	3.28	[73]
citral		64.00 ± 7.07	3rd	3.17	[73]
carvone		84.00 ± 5.09	3rd	2.27	[73]
linalool		90.00 ± 3.16	3rd	3.28	[73]

All values marked with an asterisk (*) were recalculated for comparison. ^1^ Reported for compounds whose LD_50_ cannot be calculated. Concentration: 3 μg compound/mg insect (3 mg/g insect). ^2^ Synthetic insecticides.

**Table 10 plants-12-00003-t010:** Lethal effects of pure compounds tested, on *Spodoptera frugiperda*, by contact toxicity, fumigant and immersion methods.

Compound	50% Lethal Concentration (LC_50_, CI)	Larval Stage	LogP	Reference
Contact toxicity per se				
Indoxacarb (Rumo 300 g a.i./L; DuPont do Brasil S.A.)	0.0009 (0.0006–0.0018) μL/cm^2^ *^1^	3rd	2.77	[79]
	0.0015 (0.0009–0.0021) μL/cm^2^ *^2^	3rd		[79]
thymol	0.255 (0.195–0.317) μL/cm^2^	3rd	3.28	[59]
α-pinene	2.5 (2.11–2.91) μL/cm^2^	3rd	4.37	[59]
Fumigant toxicity				
camphene	0.00067 μL/L *	2nd	4.37	[84]
α-pinene	0.0066 (0.0056–0.0079) μL/L *	2nd	4.37	[84]
β-pinene	0.016 (0.011–0.032) μL/L *	2nd	4.37	[84]
Immersion				
Geraniol	3793 (173–1281) ppm	2nd ^3^	3.28	[88]

All values marked with an asterisk (*) were recalculated for comparison. ^1^ Use Cry1A.105- and Cry2Ab-susceptible strain of *S. frugiperda*. ^2^ Use Cry1A.105- and Cry2Ab-resistant strain of *S. frugiperda*. ^3^ Larva 72 h old.

**Table 11 plants-12-00003-t011:** Principal components of the most toxic EOs against *Spodoptera frugiperda*.

Essential Oil	Main Compounds (Relative Percentage)			Reference
Ingestion				
*Cymbopogon citratus*	Geranial (47.53%)	Neral (32.5%)	nd	[71]
*Citrus limon*	Limonene * (nd)	nd	nd	[81]
*Ageratum conyzoides*	Precocene (87%)	β- caryophyllene (7.1%)	α-humulene (1.2%)	[80]
Fumigation				
*Piper septuplinervium*	α-pinene * (21%)	β-pinene * (13.8%)	Citronellal * (10.3%)	[84]
Topical application				
*Ocimum gratissimum*	Thymol * (33.2%)	p-cymene (22.5%)	γ-terpinene (21%)	[59]
Contact toxicity				
*Siparuna guianesis*	β-myrcene (74.94%)	2-undecanone (9.36%)	bicyclo-germacrene (1.52%)	[79]
Immersion				
*Piper marginatum*	Exalatacin (9.12%)	α-pinene * (8.45%)	α-phellandrene (6.97%)	[75]

All compounds marked with an asterisk (*) were tested in their pure form. nd: non determined.

**Table 12 plants-12-00003-t012:** Percentage composition of oxygenated and non-oxygenated terpenes of the most toxic EOs for *Spodoptera frugiperda*.

Essential Oils	Non-Oxygenated Terpenes (%)	Oxygenated Terpenes (%)	Reference
Ingestion			
*Ageratum conyzoides*	98.2	nd	[80]
*Cymbopogon citratus*	nd	79.03	[71]
Fumigation			
*Piper septulinervium*	81.4	11.7	[84]
Topical application			
*Ocimum gratissimum*	60.2	37.6	[59]
Contact toxicity			
*Siparuna guianesis*	80.83	10.39	[79]
Immersion			
*Piper marginatum*	52.72	18.35	[75]

The complete composition of *Citrus limon* EO was not explained in the article, so the percentage of oxygenated and non-oxygenated terpenes could not be determined. nd: non determined.

## Data Availability

Publicly available datasets were analyzed in this study. This data can be found here: https://docs.google.com/spreadsheets/d/1JaSsklncPjHAbEKGA44vTujuP165K1TE/edit?usp=sharing&ouid=101680915279321570175&rtpof=true&sd=true, accessed on 25 October 2022.

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
