# Peer review of "Can Essential Oils Be a Natural Alternative for the Control of Spodoptera frugiperda? A Review of Toxicity Methods and Their Modes of Action"

_plants, 2022, doi:10.3390/plants12010003_

Round 1

Reviewer 1 Report

Usseglio et al. present a review of the toxicity of essential oils (EO) to the fall armyworm, Spodoptera frugiperda. This is a major invasive pest in Africa, India, China and recently Australia. The review would be better placed in the MDPI journal Insects, but may be of interest to readers of Plants.
Unfortunately, the review is flawed and I cannot recommend it be accepted for publication.
Main points
(I) The authors attempt to make comparisons based on a small selection of studies (appears to be 27 publications between 2009 and 2022). Why this time-period was selected is not explained. These studies involved different larval instars (neonates, 2nd or 3rd instars). These stages differ in their susceptibility to insecticides, so the ability to draw useful comparisons is severely limited. But more importantly, the authors do not disclose the stages tested in any of their tables or figures, so we are left wondering.
(II) The authors also attempt to compare methods of exposure (topical, ingestion, fumigation), but with such a small selection of studies and a large number of different types of EOs tested (in combination with the diversity of larval stages tested), the ability to draw useful comparisons is similarly very restricted. Again, key information on methodologies is not presented.
(III) There are very few insights in this review. The authors point out that aspects of the study of EOs require additional attention, but they generally fail to draw useful conclusions or insights from the available information. Their focus on S. frugiperda to the exclusion of other species of Spodoptera, or even other noctuids, does not help their ability to provide a useful overview.
(IV) There seems to be a marked variation in their target audience. They go to great lengths to explain how synapses function, with several figures to explain this. On the other hand they expect their audience to understand advanced analytical chemistry with phrases such as the role of "Kier-Hall connectivity lag 5-non-standardized (minimum)-distance count order 4-hydrogen filled graph-total index, and the lower values of autocorrelation lag 1-non-standardized (standard deviation)-molar refractivity-hydrogen filled graph-total index", which was indecipherable for me at least.
(V) The potential of EOs as insecticides has already been widely reviewed (as the authors point out). The authors state that the novelty of their review lies in their focus on toxicity, but they do not address toxic effects that do not result in mortality at any point. Given that sublethal toxic effects can include sterility, reduced fecundity, reduced longevity, or delayed development, all of which have important implications for pest control, this is another deficiency of the review.
(VI) Several of the figures appear to be copyrighted. Do the authors have permission to reproduce these figures? I was mainly concerned by Fig 1, Fig 5 (copyrighted by Bernard Dery 2005-2016), and Fig 7 (taken from a book with an ISSN number).
(VII) The text is full of errors, omissions (of important information), contradictions (especially in percentage values) and inconsistencies. At several points, information is stated in the text and then reproduced in a figure, making the figure redundant (see my comments below). The use of percentages based on small (unknown) sample sizes makes it hard to understand the importance of the information being transmitted. I have read the text carefully and have written a total of 88 numbered points that require attention on a scanned copy of the manuscript.
Numbered points
1. Key words should not repeat words in the title.
2. In English, North, South and Central America are referred to as "the Americas".
3. S. frugiperda seems to have invaded Africa through global trade routes. Climate change has not been implicated as a major factor in its spread as far as I know.
4. It has now spread to India, China and Australia.
5. Obvious. The adult stage of ALL animals is the reproductive stage.
6. I believe this figure is copyrighted. Do you have permission to reproduce it (even in a slightly modified form)?
7. GM maize is not Bt-resistant; it is insect resistant. Suggest you use the term "Bt-maize".
8. Already stated. Delete.
9. This term is associated with popular science. What do you mean by "ecofriendly"?
10. Why 2009? What happened prior to this to merit exclusion from the dataset?
11. What do you mean "comparative effects". What was being compared?
12. 50% of how many? You need to state sample sizes when using percentage values. This is a problem throughout the manuscript.
13. Why present percentages? Are these based on the 27 articles mentioned on line 69?
14. How did you calculate 2% if the sample size was 27 articles? Unclear. (see point 12).
15. If Lippia + Ocimum together comprised 30% of the studies, why is Lamiaceae shown as 21% in Table 1? Unclear.
16. These are very small percentages, better to present the number of studies of each species, not %.
17. The text is too small to read when printed.
18. Table 2. Again, percentage values cannot be understood with reference to sample size.
19. Why not shown the 19 genera in the Table?
20. One species seems to be missing in Fig. 2.
21. Why present species (Fig 2) then genera (table 2)? Should be the other way around shouldn't it?
22. SM2. Should be Table S2. Again, what is the rationale for presenting percentages? Does "principal compound" mean the active compounds or the most abundant compounds?
23. In general you use uppercase letters for EOs, but I don't know why. For me, these compounds should be written in lowercase.
24. Why "and/or"?
25. What is the point of Figure 3? All this information is presented in Figure 4. Remove Fig. 3.
26. Indicate the sample size on which your percentage values are based.
27. Ditto.
28. Why is deltamethrin (lowercase) reported as 30% in text and 12.5% in Table 3?
29. Azamax is a product name for azadirachtin (+ other compounds), so make this clear or group it with Neem in Table 3.
30. Are you assuming that absorption occurs through the cuticle? - - if so, what are the cuticular properties that favor penetration of lipophilic compounds?
31. Unclear. Reword.
32. Figure 5 seems to be copyrighted by Bernard Dery - - it says so on the website. Do you have permission to reproduce this figure?
33. Why show an image of Manduca when you are reviewing Spodoptera frugiperda? They are not even the same family of Lepidoptera.
34. Do you mean Toxicological methods used to study S. frugiperda?
35. The term "Dorsal topication" is one I have not come across before. I suggest that topical application would be a more suitable term. Please correct the entire manuscript.
36. Again, it is not possible to understand the importance of percentage values in the absence of sample size.
37. You have already presented this information in the text (lines 166-168), so the figure is redundant and should be removed.
38. Do caterpillars have a mesothorax? Are you referring to the second thoracic segment to which the second true leg is attached? Please clarify.
39. If 21% of tests involved exposure to lethal concentrations (line 167-8), how were 89% of tests based on lethal doses?
The authors seem to confuse dose and concentration at several points in the manuscript.
40. This figure appears in a book with ISSN 2250-5350. Do you have permission to reuse it?
41. Giving abbreviations without the full name makes it very difficult to find the references. This is an issue in several of the cited references.
42. How were values transformed? This needs to be explained (applies to several of the subsequent Tables).
43a. Table 4. EOs placed on diet or mixed in with the diet?
43b. How were values "recalculated". What did this involve?
44. Values above given in ppm, but others given in mg/mL although these could easily be converted to ppm.
45a. Table 4. Please indicate the limits of each of the Application methods - I was unsure which rows corresponded to which type of application.
45b. I did not understand this point. Please rephrase.
46a. This is because there was practically NO mortality response (0-10%) observed in the experiments involving C. sinensis, correct? This problem was not one of non-dose dependence, but absence of toxicity at the concentrations tested.
46b. Please indicate the units here microliters per liter of what? Gas? Air?
47. It is well known that plant parts differ in their content of secondary chemicals; this is not just a finding for P. subtomentosum. You should cite additional articles or a review paper that show this.
48. But did they use insects of the same age or stage?
49. You mean the plant species?
50. But was this difference significant? Did 95% C.I. overlap?
51. Please explain how this information would be useful for designing toxicity tests?
52. What do values in parentheses indicate (not mentioned).
53. Should read "...for susceptible and Bt-resistant strains, respectively" ; delete following sentence.
54. Presumably there is a genetic basis and possibly a trade-off in Bt resistance against other life traits. Please explain why you think testing Bt resistance is more interesting than resistance to other insecticides.
55. ppm is not a lethal dose. This is a concentration. (see point 39).
56. See point 35 (for the entire manuscript).
57. Why present percentages when your sample size is 37? Better to say 20 were natural and 17 were synthetic insecticides.
Fig. 8 - all the information is already presented in the text, so no need to use a figure. Delete it.
58. But was this a significant difference? What do you mean adaption to indoxacarb? You mean partial resistance?
59. Was this pure indoxacarb? Or a formulated commercial product?
60. Table 9. Decis is deltamethrin.
61. What compound does "commercial product" refer to?
62. What are these values? SE? SD? 95%C.I.?
63. See point 29.
64. Why present percentages based on just 25 compounds?
65. A 2.5-fold difference in LD50 values is not much in my opinion. This would often be seen testing different insect colonies or even the same insect colony at different times of the year. Are you saying this difference is important? Did they test exactly the same larval instar?
66. If the values were significantly different, they are not really "similar", are they?
67. So what do you conclude from this? That EO compounds are about 1000 to 100,000-fold LESS toxic than most insecticides (based on Table 9). This is not convincing evidence that that could replace synthetic insecticides, even under "optimum" laboratory conditions.
68. should read "one of the most toxic EOs by ingestion"?
69. I could not understand this, please reword.
70. Extrapolated, how? Please elaborate.
71. Table 10. What do the percentage values indicate? Please explain.
72. What is a percentage analysis of functional groups? What does nd mean?
73. Please explain Log P and polar surface parameters - - readers on Plants may not be aware of these parameters of what they signify.
74. Very interesting. Please explain what  centered Broto-Moreau autocorrelation-lag 5/weighed by mass means. Also, how does the sum of E-States for (strong) hydrogen bond acceptors play a role and of course, the function of [CH] [CH3] fragments.
75. Unless you believe that the readers of Plants have a deep knowledge of analytic chemistry techniques, I think you will have to explain the importance of "Kier-Hall connectivity lag 5-non-standardized (minimum)-distance count order 4-hydrogen filled graph-total index, and the lower values of autocorrelation lag 1-non-standardized (standard deviation)-molar refractivity-hydrogen filled graph-total index." I for one will be interested to know.
76. Define QASR?
77. All these effects are likely to contribute to toxicity correct. But how important are they compared to neurotoxicity, which you focus on in the following sections?
78. What is AChE 650? I Googled this without success.
79. Define IC50.
80. Fig 10 is not informative and is the previous figure with minor modification. The text is clear enough.  Delete Fig 10.
81. Increase penetration of what?
82. Why is gene expression of GABA-T an issue of interest? Please explain.
83. Fig 12 - repetitive and not informative. Delete. (see point 80).
84. More explanation is required here. You indicate that they are important effects but why is that?
85. You should be upfront and state that all the EO studies were performed under optimum conditions in the laboratory. The field crop is a very different environment and will require careful formation technology to overcome issues of stability in storage and persistence following application.
Nowhere in the review do you address issues of the COST of EOs. (!).
86. What do you mean by "multiple studies"? Explain.
87. Why shouldn't risk assessment studies be performed on non-target organisms? You definitely need to justify such a bold assertion.
88. Numerous typos, formatting issues and missing information in the references section (I only looked at the first 10 refs).

Author Response

Response to Comments:

Reviewer #1

Main points:

  1. The authors attempt to make comparisons based on a small selection of studies (appears to be 27 publications between 2009 and 2022). Why this time-period was selected is not explained. These studies involved different larval instars (neonates, 2nd or 3rd instars). These stages differ in their susceptibility to insecticides, so the ability to draw useful comparisons is severely limited. But more importantly, the authors do not disclose the stages tested in any of their Tables or figures, so we are left wondering.

Response: Thank you for your comment. This information could be confuse. According to our survey, we only found manuscripts between 2009 and 2022 that met the selection criteria (described below) to be included in this review. To avoid confusion, we have removed the period of years from the manuscript. Besides, the selection criteria to include articles in our analysis, were highlighted in the manuscript. We agree with reviewer 1 that the effectiveness of the essential oils depends on the larval stage, and therefore the larval stage should be taken into account to compare the effectiveness of the essential oils in toxicity tests. According to the comments, we have included the larval stages used in the Tables 5 (Line 235), 6 (Line 253), 7 (Line 269), 8 (Line 284), 9 (Line 337) and 10 (Line 352),  

The selection criteria to include articles in our analysis were: 1) Research articles had to be published in scientific journals; (2) EOs had to be obtained by cold pressing or hydrodistillation. Solvent extracts were not considered; 3) Only Mortality data were considered. The effect on Reproductive, development or feeding parameters were not included in our analsys; 4) Toxicity data on cell cultures were not considered; 5) Essential oils evaluated as nanoformulations or nanoencapsulates were not included in our analysis. This information was included into the manuscript (Lines 70-76)

  1. The authors also attempt to compare methods of exposure (topical, ingestion, fumigation), but with such a small selection of studies and a large number of different types of EOs tested (in combination with the diversity of larval stages tested), the ability to draw useful comparisons is similarly very restricted. Again, key information on methodologies is not presented.

Response: We agree with the comment. However, we believe that this is precisely something to highlight. Although our survey found only 27 articles to be included in our analysis, the joint analysis of these articles yielded very interesting information to be considered in future studies on the effect of EOs against S. frugiperda. For example, there is variability in some parameters of the toxicity methods used, which makes it difficult to compare results between different authors. For this reason, we have highlighted the parameters that should be standardized (Lines 286-299). In addition, an extensive suggestion of the conditions, methodologies, concentrations, larval stages and ways of expressing the results were added into the “6. Final Considerations” section, in order to propose a standardization of the topical application method. In addition, the analysis of the available bibliography has allowed proposing the evaluation of some EOs and their main components, which have been effective on other insects but have not yet been evaluated on S. frugiperda.

We believe that the information reported in this review can serve as a guide for future studies on the effect of EOs against S. frugiperda.

  1. There are very few insights in this review. The authors point out that aspects of the study of EOs require additional attention, but they generally fail to draw useful conclusions or insights from the available information. Their focus on frugiperda to the exclusion of other species of Spodoptera, or even other noctuids, does not help their ability to provide a useful overview.

Response: Thank you for your observation. There are are many works that studied the effect of EOs against other species of Spodoptera. Therefore, we had decided to focus on S. frugiperda. However, we have added information about the effect of the EOs against S. exigua, S. littoralis and S. litura in the improve manuscript, as suggested. These information is shown in SM1, and discussed in the subsection titled “Comparison of insecticidal effects of EOs between S. frugiperda and the S. littoralis-S. litura-S. exigua complex”.

Although our survey found only 27 articles to be included in our analysis, the joint analysis of these articles yielded very interesting information to be considered in future studies on the effect of EOs against S. frugiperda. For example, there is variability in some parameters of the toxicity methods used, which makes it difficult to compare results between different authors. For this reason, we have highlighted the parameters that should be standardized (Lines 286-299). In addition, an extensive suggestion of the conditions, methodologies, concentrations, larval stages and ways of expressing the results were added into the “Conclusion” section, in order to propose a standardization of the topical application method. In addition, the analysis of the available bibliography has allowed proposing the evaluation of some EOs and their main components, which have been effective on other insects but have not yet been evaluated on S. frugiperda.

We believe that the information reported in this review can serve as a guide for future studies on the effect of EOs against S. frugiperda.

  1. There seems to be a marked variation in their target audience. They go to great lengths to explain how synapses function, with several figures to explain this. On the other hand they expect their audience to understand advanced analytical chemistry with phrases such as the role of "Kier-Hall connectivity lag 5-non-standardized (minimum)-distance count order 4-hydrogen filled graph-total index, and the lower values of autocorrelation lag 1-non-standardized (standard deviation)-molar refractivity-hydrogen filled graph-total index", which was indecipherable for me at least.

Response: Thank you for your comments. According with your suggestion We have modified the section "5.3 Structure-activity relationship" to favor the understanding of the analyzed parameters. (Lines 440-494).

  1. The potential of EOs as insecticides has already been widely reviewed (as the authors point out). The authors state that the novelty of their review lies in their focus on toxicity, but they do not address toxic effects that do not result in mortality at any point. Given that sublethal toxic effects can include sterility, reduced fecundity, reduced longevity, or delayed development, all of which have important implications for pest control, this is another deficiency of the review.

Response: We agree with the comment. There are very interesting information about the sublethal toxic effects of EOs on sterility, fecundity, longevity, or delayed development, which could have implications for pest control. However, it was very difficult to cover all these issues in this review, and therefore we have focused on toxicity studies. The importance of these issues, and the need to review them in the future, has been added in the “Introduction” section (Lines 67-70).

  1. Several of the figures appear to be copyrighted. Do the authors have permission to reproduce these figures? I was mainly concerned by Fig 1, Fig 5 (copyrighted by Bernard Dery 2005-2016), and Fig 7 (taken from a book with an ISSN number).

Response: Thank you for your observation. We believed that they were figures of free access. To avoid any mistake, we have changed these figures.

  1. The text is full of errors, omissions (of important information), contradictions (especially in percentage values) and inconsistencies. At several points, information is stated in the text and then reproduced in a figure, making the figure redundant (see my comments below). The use of percentages based on small (unknown) sample sizes makes it hard to understand the importance of the information being transmitted. I have read the text carefully and have written a total of 88 numbered points that require attention on a scanned copy of the manuscript.

Response: We want to thank you for the effort you have made to review our article. All your comments and suggestions have helped to improve it in a superlative way. we have revised the manuscript taking into account all your comments. The response to each of these numbered points is detailed below.

  1. Key words should not repeat words in the title.

Response: Thank you for your observation. The Key words were replaced for the following: “Natural insecticides, insect pest, phytochemicals, mortality rate” (Line 24)

  1. In English, North, South and Central America are referred to as "the Americas".

Response: According to the comment, it was included in the manuscript (Lines 35-37).

  1. frugiperda seems to have invaded Africa through global trade routes. Climate change has not been implicated as a major factor in its spread as far as I know.

Response: According to the comment, this sentence was modify as follow: “In addition, as a result of the expansion of the agricultural frontiers, it is now considered to be an invasive pest in African countries, China, India and Australia.” (Lines 37-39).

  1. It has now spread to India, China and Australia.

Response: Thank you for your observation. We have included India and Australia to the sentence. This sentence has change to: “In addition, as a result of the expansion of the agricultural frontiers, it is now considered to be an invasive pest in African countries, China, India and Australia.” (Lines 37-39).

  1. The adult stage of ALL animals is the reproductive stage.

Response: Thank you for your observation. This obviousness regarding the reproductive stage was removed from the manuscript (Line 39).

  1. I believe this figure is copyrighted. Do you have permission to reproduce it (even in a slightly modified form)?

Response: Thank you very much for this warning. We thought that it is as a“Creative Commons License”. However, to avoid any mistake, we have eliminated this figure from the manuscript. 

  1. GM maize is not Bt-resistant; it is insect resistant. Suggest you use the term "Bt-maize".

Response: Thank you for your suggestion. The term “Bt-resistant maize” was changed to “Bt-maize” into the manuscript (Lines 45 and 46).

  1. Already stated. Delete.

Response: This phrase was removed to the manuscript, as suggested.

  1. This term is associated with popular science. What do you mean by "ecofriendly"?

Response: The term "ecofriendly" refers to natural alternatives to synthetic pesticides that are less harmful to the environment and health. This term has been incorporated in some scientific articles[1–10]. So, we considered it appropriate to use it in our manuscript. However, if Plant´s editor deems the use of this term inappropriate, we will remove it from the manuscript.

References:

  1. Gupta, S.; Dikshit, A.K. Biopesticides: An Ecofriendly Approach for Pest Control. J. Biopestic. 2010, 3, 186–188.
  2. Araniti, F.; Landi, M.; Laudicina, V.A.; Abenavoli, M.R. Secondary Metabolites and Eco-Friendly Techniques for Agricultural Weed/Pest Management. Plants 2021, 10, 12–15, doi:10.3390/plants10071418.
  3. Bhan, S.; Mohan, L.; Srivastava, C.N. Nanopesticides: A Recent Novel Ecofriendly Approach in Insect Pest Management. J. Entomol. Res. 2018, 42, 263–270, doi:10.5958/0974-4576.2018.00044.0.
  4. El-Saadony, M.T.; Abd El-Hack, M.E.; Taha, A.E.; Fouda, M.M.G.; Ajarem, J.S.; Maodaa, S.N.; Allam, A.A.; Elshaer, N. Ecofriendly Synthesis and Insecticidal Application of Copper Nanoparticles against the Storage Pest Tribolium Castaneum. Nanomaterials 2020, 10, doi:10.3390/nano10030587.
  5. Srivastava, S.; Mishra, H.N. Ecofriendly Nonchemical/Nonthermal Methods for Disinfestation and Control of Pest/Fungal Infestation during Storage of Major Important Cereal Grains: A Review. Food Front. 2021, 2, 93–105, doi:10.1002/fft2.69.
  6. Jesser, E.; Yeguerman, C.; Stefanazzi, N.; Gomez, R.; Murray, A.P.; Ferrero, A.A.; Werdin-González, J.O. Ecofriendly Approach for the Control of a Common Insect Pest in the Food Industry, Combining Polymeric Nanoparticles and Post-Application Temperatures. J. Agric. Food Chem. 2020, 68, 5951–5958, doi:10.1021/acs.jafc.9b06604.
  7. Chaudhari, A.K.; Singh, V.K.; Kedia, A.; Das, S.; Dubey, N.K. Essential Oils and Their Bioactive Compounds as Eco-Friendly Novel Green Pesticides for Management of Storage Insect Pests: Prospects and Retrospects. Environ. Sci. Pollut. Res. 2021, 28, 18918–18940, doi:10.1007/s11356-021-12841-w.
  8. Kumar, R.; Mishra, A.K.; Dubey, N.K.; Tripathi, Y.B. Evaluation of Chenopodium ambrosioides Oil as a Potential Source of Antifungal, Antiaflatoxigenic and Antioxidant Activity. Int. J. Food Microbiol. 2007, 115, 159–164, doi:10.1016/j.ijfoodmicro.2006.10.017.
  9. Singh, V.P.; Singh, A. Pesticides in Crop Production: Physiological and Biochemical Action; 2020; ISBN 9781119432197.
  10. Muhammad, A.; Kashere, M.A. NEEM, Azadirachta Indica L. (A. Juss): An eco-friendly botanical insecticide for managing farmers’ insects pest problems - a review. FUDMA J. Sci. 2021, 4, 484–491, doi:10.33003/fjs-2020-0404-506.

  1. Why 2009? What happened prior to this to merit exclusion from the dataset?

Response: Thank you for this observation. According to our survey, we only found manuscripts between 2009 and 2022 that met the selection criteria to be included in this review. To avoid confusion, we have removed the period of years from the manuscript (Lines 61-62).

  1. What do you mean "comparative effects". What was being compared?

Response: Thank you for your inquiry. When dealing with articles which used diverse methods, concentration units (µl/l, mM, % µl/cm2) or the way of expressing mortality (mortality percentage, lethal dose 50, lethal concentration 50), the results could not be always compared. For this reason, we used reference units (the most cited for the methods used, for example mg/g insect in the topical application method) and we converted the results, when possible, to obtain doses in equal units, and so that they can be compared between articles.

  1. 50% of how many? You need to state sample sizes when using percentage values. This is a problem throughout the manuscript.

Response: Thank you for your suggestion. This percentage refers to the 27 articles that were selected according to the selection criteria. It was clarified in the manuscript as follow “More frequently than 50% of the 27 selected articles referred to EOs obtained from just three plant families: Piperaceae, Lamiaceae and Verbenaceae, in order of decreasing frequency.” (Lines 87-89).

  1. Why present percentages? Are these based on the 27 articles mentioned on line 69?

Response: We present the results as percentages because we believe that it facilitates the analysis of the results for the reader. Nevertheless, we change the percentages for the number of occurrences of plant family in the 27 articles.

  1. How did you calculate 2% if the sample size was 27 articles? Unclear. (see point 12).

Response: We calculated these percentages based on the total of number of occurrences of plant families (56) in the analyzed articles. Nevertheless, to avoid confusion we change the percentages for the number of occurrences of plant family (Lines 88-90, Table 1).

  1. If Lippia + Ocimum together comprised 30% of the studies, why is Lamiaceae shown as 21% in Table 1? Unclear.

Response: These percentages were calculated based on the 19 plant genera studied in the selected articles. Nevertheless, to avoid confusion we change the percentages for the number of occurrences of plant genera (Lines 98-99, Table 2)

  1. These are very small percentages, better to present the number of studies of each species, not %.

Response: Thank you for your suggestion. We have replaced figure 2 with a new Table (Table 3) which also report the number of studies of each species (Line 108).

  1. The text is too small to read when printed.

Response: Thank you for your observation The figure 2 was replaced by a new Table (Table 3), and this problem was solved.

  1. Table 2. Again, percentage values cannot be understood with reference to sample size.

Response: These percentages were calculated based on the 19 plant genera studied in the selected articles. The clarification is shown at the footnote of Table 2. (Line 108).

  1. Why not shown the 19 genera in the Table?

Response: We have not included the 19 genera in the Table so that it is not very extensive. The genera not included in the Table were indicated at the footnote (Lines 101-106).

  1. One species seems to be missing in Fig. 2.

Response: The figure 2 was replaced by a new Table (Table 3), and all the species were included.

  1. Why present species (Fig 2) then genera (Table 2)? Should be the other way around shouldn't it?

Response: Thank you for this comment. Following all the comments, we have replaced the Fig 2 with a new Table (Table 3), in the improve version of the manuscript. Therefore, plant genera are presented before species in the improved manuscript.

  1. Should be Table S2. Again, what is the rationale for presenting percentages? Does "principal compound" mean the active compounds or the most abundant compounds?

Response: Thank you for your suggestion. The title of the Table was changed to S15, as suggested and due to the addition of new supplementary material. The term “principal compound” refers to the most abundant compounds. This information is shown at the footnote of the Table.

We decided to use percentages to make it easier for the reader to analyze the results. However, if the editor considers it appropriate, we can change the percentages by the number of articles which each of the VOCs is mentioned as the main component of each EO.

  1. In general you use uppercase letters for EOs, but I don't know why. For me, these compounds should be written in lowercase.

Response: According to the comment, all the compounds were written in lowercase.

  1. Why "and/or"?

Response: It was modified (Lines 112-113).

  1. What is the point of Figure 3? All this information is presented in Figure 4. Remove Fig. 3.

Response:  We agree with the comment, and the figure 3 was removed, as suggested.

  1. Indicate the sample size on which your percentage values are based.

Response: The percentages were calculated based on 23% of articles (of 27 selected) that test pure VOCs. It was clarified in the manuscript (Lines 112-114, page 5).

  1.  

Response: Percentages calculated based on 41.6% of the articles (of 27 selected) that use commercial insecticides as positive controls (Lines 131-143).

  1. Why is deltamethrin (lowercase) reported as 30% in text and 12.5% in Table 3?

Response: Thank you for this observation. It was a typographical mistake. The correct value was reported in Table 4 (12.5%). So, the phrase in text was changed to: “Deltamethrin (12.5%), a synthetic pyrethroid pesticide used in livestock, aquaculture and agriculture due to its low residue and high toxicity, as well as its great efficacy, was the most frequently used synthetic insecticide as the positive control” (Lines 134-137).

  1. Azamax is a product name for azadirachtin (+ other compounds), so make this clear or group it with Neem in Table 3.

Response: Thank you for your suggestion. This information was added into the Table 4, as suggested (Lines 139-141).

  1. Are you assuming that absorption occurs through the cuticle? - - if so, what are the cuticular properties that favor penetration of lipophilic compounds?

Response: According to the comment, an explanation about the cuticular properties that favor the penetration of lipophilic compounds was added to the manuscript as follow: “It is widely known that organophosphate insecticides, such as Dichlorvos (DDVP), penetrate through the integument until they reach the hemolymph and, subsequently, their site of action[47,48]. In turn, there is a correlation between resistance to insecticides and cuticular penetration[49–51]. The non-polar nature of the insect cuticle, composed mainly of aliphatic hydrocarbons, chitin and waxes, would favor the entry of lipophilic compounds, such as those present in EOs[35,42,49,52]. This is crucial when choosing the method used to assess the toxicity of EOs on S. frugiperda.” (Lines 152-158).

  1. Reword.

Response: According to the comment, this phrase was changed to: “It is a critical property to be considered when choosing the method to assess the toxicity of EOs on S. frugiperda” (Lines 157-158).

  1. Figure 5 seems to be copyrighted by Bernard Dery - - it says so on the website. Do you have permission to reproduce this figure?

Response: Thank you very much for this warning. To avoid any mistake, the Figure 5 has been removed, and a new schematic figure made by us was added as Figure 2 (Lines 170-176)

  1. Why show an image of Manduca when you are reviewing Spodoptera frugiperda? They are not even the same family of Lepidoptera.

Response: It had been chosen as a standard lepidoterora where the body segments and the spiracles were clearly distinguished. To avoid confusion, this image has been changed to another more representative of S. frugiperda (Lines 170-176).

  1. Do you mean Toxicological methods used to study frugiperda?

Response: Thank you for your observation. The title was changed in the manuscript to “Toxicological methods against Spodoptera frugiperda.” (Line 177)

  1. The term "Dorsal topication" is one I have not come across before. I suggest that topical application would be a more suiTable term. Please correct the entire manuscript.

Response: Thank you for your suggestion. The term “Dorsal Topication” were replaced for “Topical application” in the manuscript.

  1. Again, it is not possible to understand the importance of percentage values in the absence of sample size.

Response: According to the comment, we have included the number of tests implemented for each toxicity method in the articles analyzed into the now figure 4 (ex figure 6). In total, 33 toxicity methods have been reported. Some articles evaluated the toxicity of EOs by means of more than one method.

  1. You have already presented this information in the text (lines 166-168), so the figure is redundant and should be removed.

Response: We have changed the information provided into the now figure 4 (ex figure 6), so it is not repeated in the text. However, if the reviewer consider it necessary, we can remove figure 6 (now Fig. 4)

  1. Do caterpillars have a mesothorax? Are you referring to the second thoracic segment to which the second true leg is attached? Please clarify.

Response: Thank you for your suggestion. Yes, it is the second thoracic segment that Romero and Navarro[1] defines as “mesothorax”. To avoid confusion, we have changed the word mesothorax to " second thoracic segment " in the manuscript (Lines 188-189).

References:

  1. Romero, F.; Navarro, F. Lepidoptera. In Macroinvertebrados bentónicos sudamericanos: sistemática y biología; Domínguez, E., Fernández, H.R., Eds.; Fundación Miguel Lillo, Tucumán, Argentina, 2009; pp. 309–340.

  1. If 21% of tests involved exposure to lethal concentrations (line 167-8), how were 89% of tests based on lethal doses? The authors seem to confuse dose and concentration at several points in the manuscript.

Response: Thank you for your observation. It could be confusing. In total, 55% (18) of the toxicity tests to study the insecticidal effects of EOs on FAW, used topical application of insects. Within these, the 89% reported the toxic results as lethal doses, while that the remaining 11% reported the toxic results as mortality (percentage). To avoid confusion, these percentages have been changed by “number of occurrences in the literature”. Therefore, this phrase was modified as follow “The methods used to study the insecticidal effects of EOs on FAW are shown in Figure 3. Thirty-three toxicity assays were registered in the analysis of the articles. Of these assays, the most cited method for studying toxicity was the topical application of insects (18), followed by toxicity by ingestion (8) and contact toxicity (4) techniques, with fumigant toxicity (2) and immersion (1) being the least used methods. It should be noted that the methodologies usually differed slightly between articles.” (Lines 178-183).

  1. This figure appears in a book with ISSN 2250-5350. Do you have permission to reuse it?

Response: Thank you very much for the warning. This figure was replaced by a schematic image made entirely by us (Line 170).

  1. Giving abbreviations without the full name makes it very difficult to find the references. This is an issue in several of the cited references.

Response: This reference was modified, as suggested (Lines 171-176). In addition, we have reviewed all the references in the manuscript.

  1. How were values transformed? This needs to be explained (applies to several of the subsequent Tables).

Response: The values were transformed in different ways depending on the toxicity method used. For the transformations different aspects were taken into account: the molecular weights of the compounds, their density, area/volume of application, whether or not they were applied directly to the insect, etc. For example, for the results of the contact toxicity experiments, the concentration was defined as: amount of compound applied (expressed in µl, converted using density when necessary) per area of application.

If the editor considers it necessary, we can add an SM where the transformations carried out for the results of each methodology are explained.

  1. Table 4. EOs placed on diet or mixed in with the diet?
  2. How were values "recalculated". What did this involve?

Response: a. The EOs were mixed with the diet. This clarification was added in the manuscript (Lines 262-265 and Table 7).

  1. The values were recalculated based on their molecular weight, density and application volume. This information was added in the manuscript

  1. Values above given in ppm, but others given in mg/mL although these could easily be converted to ppm.

Response: The values were converted to ppm, as suggested (Table 7).

  1. Table 4. Please indicate the limits of each of the Application methods - I was unsure which rows corresponded to which type of application.
  2. I did not understand this point. Please rephrase.

Response: a. According to the comment, a dividing line was added between the different methods in Table 4, now Table 7 (Line 269, Table 7).

  1. To avoid confusion, this phrase was changed to: “(…) and Citrus limon showed the higher toxic effect when the EO was mixed with the artificial diet (98.29 ppm)”. (Lines 263-265)

  1. This is because there was practically NO mortality response (0-10%) observed in the experiments involving C. sinensis, correct? This problem was not one of non-dose dependence, but absence of toxicity at the concentrations tested.
  2. Please indicate the units here microliters per liter of what? Gas? Air?

Response: a. Yes, it is right. The EO of C. sinensis did not show mortality affect against S. frugiperda. So, the lethal concentration could not be calculated. According to the comment, the sentence was changed to “As a non-toxicity effect was observed in the artificial diet mixed with EO of C. sinensis, its LC50 could not be calculated” (Lines 265-266).

  1. According to the comment, this information was added into the manuscript.

  1. It is well known that plant parts differ in their content of secondary chemicals; this is not just a finding for subtomentosum. You should cite additional articles or a review paper that show this.

Response: We have included new references related with the phytochemical variations between the different plant organs (Lines 278-280).

Németh-Zámbori, É. Natural Variability of Essential Oil Components. In Handbook of Essential Oils; Husnu Can Baser, K., Buchbauer, G., Eds.; Taylor & Francis Group: Boca Ratón, 2020; p. 40 ISBN 9781351246460.

Ilardi, V.; Badalamenti, N.; Bruno, M. Chemical Composition of the Essential Oil from Different Vegetative Parts of Foeniculum Vulgare Subsp. Piperitum (Ucria) Coutinho (Umbelliferae) Growing Wild in Sicily. Nat. Prod. Res. 2022, 36, 3587–3597, doi:10.1080/14786419.2020.1870227.

Chalchat, J.C.; Garry, R.P.; Muhayimana, A. Essential Oil of Tagetes Minuta from Rwanda and France: Chemical Composition According to Harvesting Location, Growth Stage and Part of Plant Extracted. J. Essent. Oil Res. 1995, 7, 375–386, doi:10.1080/10412905.1995.9698544.

  1. But did they use insects of the same age or stage?

Response: In both articles, the toxic effect of the EO of Ocimum gratissimum was evaluated by topical application in the third-stage larvae. The difference obtained in lethal doses could be explained by differences in plant varieties used. According to the comment, we have included a new column in the Tables indicating the larval stage used in each test. It would allow the reader to carry out the same analysis that you have carried out.

  1. You mean the plant species?

Response: Thanks to your observation. Yes, we mean the plant species. It was clarified in the manuscript (Lines 222-224).

  1. But was this difference significant? Did 95% C.I. overlap?

Response: Thank you for your observation. At 48 and 96 h no statistically, significant difference was observed between the LD50 obtained, since the CI overlap. However, at 24 hours there was statistically significant difference with the LD50 calculated for 48 and 96 hours. According to your observation, we have modified the information provided in the manuscript as follows: “The lethal dose calculated at 48 h after application was the most toxic (LD50 = 3.39 mg/g insect), followed by the dose at 96 h (LD50 = 3.56 mg/g insect), despite not presenting significant differences with the LD50 determined for 48 h. The lethal dose calculated at 24h being the least active (LD50 = 4.62 mg/g insect) and statistically different from the LD50 obtained for 48 and 96 h.” (Lines 229-233).

  1. Please explain how this information would be useful for designing toxicity tests?

Response: In our study, we have identified a great difference in the methods used to evaluate the toxicity of EOs against S. frugiperda. Slightly differences in some parameters of the methods used can modify the toxicity values. Hence, we believe that some parameters should be unified to make more comparable the toxicity values reported in different works. For this reason, we have highlighted the parameters that should be standardized (Lines 286-299). In addition, an extensive suggestion of the conditions, methodologies, concentrations, larval stages and ways of expressing the results were added into the “Conclusion” section, in order to propose a standardization of the topical application method (Lines 693-706).

  1. What do values in parentheses indicate (not mentioned).

Response: Thank you for the comment. The values in parentheses indicate the CI. This information was added into the manuscript (Table 5).

  1. Should read "...for susceptible and Bt-resistant strains, respectively" ; delete following sentence.

Response: The sentence was deleted, as suggested.

  1. Presumably there is a genetic basis and possibly a trade-off in Bt resistance against other life traits. Please explain why you think testing Bt resistance is more interesting than resistance to other insecticides.

Response: Thank you for your observation. We do not consider that resistance to Bt is more interesting than resistance to other insecticides. However, it is an emerging resistance, of which there were no records before 2015, we consider that it is also important to take it into account when evaluating the methods alternative control chemicals since the larvae may present physiological changes between those resistant or not to Bt that may affect the effectiveness of EO, for example. A brief clarification of the importance of considering Bt resistance has been included in the manuscript (Lines 245-251).

  1. ppm is not a lethal dose. This is a concentration. (see point 39).

Response: According to the comment, the sentence was modified (Lines 259-262).

  1. See point 35 (for the entire manuscript).

Response: These percentages were calculated based on the 37 pure compounds tested.

  1. Why present percentages when your sample size is 37? Better to say 20 were natural and 17 were synthetic insecticides.

Response: According to the comment, it was modified, as suggested (Lines 313-315).

  1. 8 - all the information is already presented in the text, so no need to use a figure. Delete it.

Response: The figure 8 was delete from the manuscript, as suggested.

  1. But was this a significant difference? What do you mean adaption to indoxacarb? You mean partial resistance?

Response: Thank you very much for your observation. There was no statistically significant difference between the lethal dose calculated for susceptible larvae and that calculated for resistant larvae, because their CIs overlap. So, the sentence was modified as follows: “Only in the case of the contact method was a synthetic insecticide tested, with a lethal dose of 9 x 10-4 µl/cm2 being obtained for Cry1A.105 and Cry2Ab susceptible strains of FAW, while for the resistant strain the lethal dose obtained was almost double this value (1.5 x 10-3 µl/cm2), however, there are no statistical differences between them.” (Lines 346-350).

  1. Was this pure indoxacarb? Or a formulated commercial product?

Response: Thank you for the comment. Indoxicarb is the active ingredient in Rumo insecticide (DuPont™). This compound is a broad-spectrum lepidoptera insecticide represents a new class of insecticides the oxidiazines.  This information was added in the manuscript. (Table 10).

  1. Table 9. Decis is deltamethrin.

Response: Yes, it is right. Decís's active ingredient is deltamethrin. We have decided to use its commercial name since that article specifies that they used the commercial formulation. In the remaining articles where they use deltamethrin, they do not indicate whether they use the pure compound or a formulated one. Including this information seemed interesting for readers when comparing the results obtained for this synthetic insecticide. However, if the editor considers it necessary, we can only report the name of the active ingredient in the Table.

  1. What compound does "commercial product" refer to?

Response: The authors of this work did not specify the main component of the product used. They refered to it as "commercial product", so we could not provide more information. If the editor considers it necessary, we can remove that row from the Table.

  1. What are these values? SE? SD? 95%C.I.?

Response: Mortality percentages are reported with their standard deviations (SD). According with your observation, this information was included in, now, Table 9.

  1. See point 29.

Response: According with the comment, the percentages were replaced by number of compounds (Lines 313-315).

  1. Why present percentages based on just 25 compounds?

Response: We believed that the use of percentages in these results could make it easier to interpret. However, these percentages were changed by the number of compounds of each type used, as suggested. (Line 313-315).

  1. A 2.5-fold difference in LD50 values is not much in my opinion. This would often be seen testing different insect colonies or even the same insect colony at different times of the year. Are you saying this difference is important? Did they test exactly the same larval instar?

Response: Thank you for your observation. Both authors used third stage larvae (this information was included in the improve Table) to test the toxic effect of linalool. In this sentence we wanted to highlight the existence of variability between the different tests using the same compound and the same larval stage. This information on variability could be useful for readers

  1. If the values were significantly different, they are not really "similar", are they?

Response: Thank you for your observation. All values reported for deltamethrin from different articles did not present statistically significant differences between them.  In all cases, their CI overlap.

  1. So what do you conclude from this? That EO compounds are about 1000 to 100,000-fold LESS toxic than most insecticides (based on Table 9). This is not convincing evidence that that could replace synthetic insecticides, even under "optimum" laboratory conditions.

Response: We agree with you. The essential oils and their main components are less effective than synthetically manufactured insecticides. However, we believe that synergistic mixtures between EOs, or their main components, with traditional synthetic insecticides could be a feasible alternative to be analyzed. We have incorporated a reflection on this topic in the new version of the manuscript (Lines 692-685).

  1. should read "one of the most toxic EOs by ingestion"?

Response: According to the comment, the sentence was modified (Lines 361-362).

  1. I could not understand this, please reword.

Response: The sentence was changed to: “In contrast, the main components of the most toxic EO by the fumigant method (P. septuplinervium), the isomers α and β pinene (Table 6), presented high toxicity values (LD50α-pinene= 0.0066 µl/l; LD50βpinene= 0.016 µl/l)[84] (Table 10), allowing the toxic fumigant effect of the EO to be associated with the presence of these two major compounds.” (Lines 364-367).

  1. Extrapolated, how? Please elaborate.

Response: Thank you for your observation. We tried to do an association between the toxic effect that was determined for EO and the effect observed for the pure compound to avoid confusion, the sentence was rewritten to: “In contrast, the main components of the most toxic EO by the fumigant method (P. septuplinervium), the isomers α and β pinene (Table 6), presented high toxicity values (LD50α-pinene= 0.0066 µl/l; LD50βpinene= 0.016 µl/l)[84] (Table 10), allowing the toxic fumigant effect of the EO to be associated with the presence of these two major compounds”(Lines 364-367).

  1. Table 10. What do the percentage values indicate? Please explain.

Response: These percentage values indicate the relative percentage of each compound in the EO. According to the comment, this information was included in the now Table 11 (Lines 378, now Table 11).

  1. What is a percentage analysis of functional groups? What does nd mean?

Response: It indicates the amount of oxygenated and non-oxygenated compounds present in the essential Oil. To avoid confusion, the sentence was modified as follows: “The percentage of oxygenated and non-oxygenated compounds present in the most toxic EOS is shown in Table 12.” (Lines 442-443).

Nd means not determined. This clarification was included in the corresponding Table footer.

  1. Please explain Log P and polar surface parameters - - readers on Plants may not be aware of these parameters of what they signify.

Response: Thank you for your comments. The definition of LogP and polar surface were included in the manuscript as follow: “For S. zeamais these studies reveal that the toxicity of the natural compounds are re-lated to descriptors such as the LogP (octanol–water partition coefficient) and the acidity (pKa), having these a key role in reaching the target site of action[93]” (Lines 466-469).

  1. Very interesting. Please explain what centered Broto-Moreau autocorrelation-lag 5/weighed by mass means. Also, how does the sum of E-States for (strong) hydrogen bond acceptors play a role and of course, the function of [CH] [CH3] fragments.

Unless you believe that the readers of Plants have a deep knowledge of analytic chemistry techniques, I think you will have to explain the importance of "Kier-Hall connectivity lag 5-non-standardized (minimum)-distance count order 4-hydrogen filled graph-total index, and the lower values of autocorrelation lag 1-non-standardized (standard deviation)-molar refractivity-hydrogen filled graph-total index." I for one will be interested to know

Response: Thank you for your comments. According with your suggestion We have modified the section "5.3 Structure-activity relationship" to favor the understanding of the analyzed parameters. (Lines 440-494).

  1. Define QASR?

Response: The QSAR (Quantitative structural-activity relationship) was added in the manuscript, as suggested (Line 462).

  1. All these effects are likely to contribute to toxicity correct. But how important are they compared to neurotoxicity, which you focus on in the following sections?

Response: Thank you for this comment. Although, the main effect of the essential oils is related to the alteration of the nervous system, others mechanisms could contribute to their toxic effect. Here, we wanted to highlight it idea. We agree with the comment, this could be confusing. To avoid confusion, these lines were removed from the manuscript.

  1. What is AChE 650? I Googled this without success.

Response: Thank you for your observation. It was a typographical mistake. The phrase was modified to “De Oliveira et al.[98] showed that both the EO of C. flexuosus and citral (its main component) inhibited FAW AChE by 450 fold more than the methomyl insecticide used as a positive control.”. (Lines 524-526).

  1. Define IC50.

Response: The definition of IC50 (Inhibitory Concentration 50) was added into the manuscript, as suggested (Line 526).

  1. Fig 10 is not informative and is the previous figure with minor modification. The text is clear enough. Delete Fig 10.

Response: According to the comment, the figure 10 was deleted.

  1. Increase penetration of what?

Response: Thank you for your observation. The presence of numerous lipophilic compounds in the EO of Salvia hispanica could favor the cuticle penetration of its main bioactive compound, α-thujone, facilitating its arrival at the target sites, and exerting a synergistic effect. This clarification was included in the new version of the manuscript (Lines 567-570).

  1. Why is gene expression of GABA-T an issue of interest? Please explain.

Response: GABA-T is the enzyme responsible for the degradation of GABA in the mitochondrial matrix, so its inhibition leads to an accumulation of this neurotransmitter. This accumulation would cause the constant excitation of the nervous system and, consequently, the death of the organism. The function of GABA-T was included in the manuscript (Lines 573-578).

  1. Fig 12 - repetitive and not informative. Delete. (see point 80).

Response: Thank you for your suggestion. According to the comment, the figure 12 was deleted.

  1. More explanation is required here. You indicate that they are important effects but why is that?

Response: Although, the main effect of the essential oils is related to the alteration of the nervous system, others mechanisms could contribute to their toxic effect. So, the design of complex formulations that in addition to affecting the nervous system induce other toxic effect such as oxidative stress or damage at the macromolecular level, should be considered. This information was added in the manuscript (Lines 618-639).

  1. You should be upfront and state that all the EO studies were performed under optimum conditions in the laboratory. The field crop is a very different environment and will require careful formation technology to overcome issues of stability in storage and persistence following application.

Response: Thank you for your suggestion. We agree with the comment. We have added the clarification that the effectiveness of EOs was determined under optimum laboratory conditions, and that their effectiveness under field conditions could be very different. Therefore, new technologies must be development to induce the stability and persistence of essential oils under field conditions. In addition, we have added information on the incorporation of EOs in nanoencapsulations that allow their controlled release (Lines 647-653).

Stamm, K.; Saltin, B.D.; Dirks, J.H. Biomechanics of Insect Cuticle: An Interdisciplinary Experimental Challenge. Appl. Phys. A Mater. Sci. Process. 2021, 127, 1–9, doi:10.1007/s00339-021-04439-3.

Dinesh, D.; Murugan, K.; Subramaniam, J.; Paulpandi, M.; Chandramohan, B.; Pavithra, K.; Anitha, J.; Vasanthakumaran, M.; Fraceto, L.F.; Wang, L.; et al. Salvia leucantha Essential Oil Encapsulated in Chitosan Nanoparticles with Toxicity and Feeding Physiology of Cotton Bollworm Helicoverpa armigera. In Biopesticides; Elsevier, 2022; pp. 159–181.

Suresh, U.; Murugan, K.; Panneerselvam, C.; Thabiani, A.; Cianfaglione, K.; Wang, L.; Maggi, F. Encapsulation of Sea Fennel (Crithmum Maritimum) Essential Oil in Nanoemulsion and SiO 2 Nanoparticles for Treatment of the Crop Pest Spodoptera litura and the Dengue Vector Aedes Aegypti. Ind. Crop. Prod. 2020, 158, 113033, doi:10.1016/j.indcrop.2020.113033.

Ibrahim, S.S.; Abou-elseoud, W.S.; Elbehery, H.H.; Hassan, M.L. Chitosan-Cellulose Nanoencapsulation Systems for Enhancing the Insecticidal Activity of Citronella Essential Oil against the Cotton Leafworm Spodoptera littoralis. Ind. Crop. Prod. 2022, 184, 115089, doi:10.1016/j.indcrop.2022.115089.

Tortorici, S.; Cimino, C.; Ricupero, M.; Musumeci, T.; Biondi, A.; Siscaro, G.; Carbone, C.; Zappala, L. Nanostructured Lipid Carriers of Essential Oils as Potential Tools for the Sustainable Control of Insect Pests. Ind. Crop. Prod. 2022, 181, doi:10.1016/j.indcrop.2022.114766.

Lemus de la Cruz, A..; Barrera-Cortés, J.; Lina-García, L.P.; Ramos-Valdivia, A.C.; Santillán, R. Nanoemulsified Formulation of Cedrela odorata Essential Oil and Its Larvicidal Effect against Spodoptera frugiperda (J.E. Smith). Molecules 2022, 27, doi:10.3390/molecules27092975.

  1. Nowhere in the review do you address issues of the COST of EOs. (!).

Response: According to the comment, the following paragraph on the cost of EOs was added to the manuscript: “Although this cost is high, there is a tendency for producers worldwide to change the cost/efficiency paradigm to one where the health of people, animals and the environment is the center[145,146]. In turn, knowing the main compounds that constitute the EOs and, in many cases, their bioactivity, one could think of synthesizing them so that the costs are lower” (Lines 659-663).

  1. What do you mean by "multiple studies"? Explain.

Response: Thank you for your observation. The word “multiple” was deleted in the new version of the manuscript.

  1. Why shouldn't risk assessment studies be performed on non-target organisms? You definitely need to justify such a bold assertion.

Response: Thank you very much for your observation. There was an error during the writing of the manuscript. We wanted to emphasize the importance of conducting studies on the effect of essential oils on non-target organisms despite the fact that they are considered safe. Thanks to your observation, we have corrected the sentence in the manuscript (Lines 654-659).

  1. Numerous typos, formatting issues and missing information in the references section (I only looked at the first 10 refs).

Response: Thank you very much for your observation. All the references were reviewed.

Reviewer 2 Report

The document es very good, and complete, just, in the abstract you writed about extraction methods ,however in tha text i dont found this information.

Tha other thing i would like to find some information about application techniques and one table about principal comounds´s log P

Author Response

Response to Comments:

Reviewer #2

  1. The document es very good, and complete, just, in the abstract you writed about extraction methods, however in tha text i dont found this information.

Response: Thank you for your observation. The extraction methods were mentioned in the lines 110-115. These methods are widely used for the extraction of EOs, so we considered that a description of them was not necessary. If the editor considers it necessary, we can include it.

  1. Tha other thing i would like to find some information about application techniques and one Table about principal comounds´s log P

Response: According to the comment, the LogP values of the pure compounds were added in Tables 9 and 10.

Reviewer 3 Report

The title is quite interesting but the authors has cite very few works related to EO insecticidal activities. To answer the question raised in the title the author need to add more literature and add comprehensive table regarding other pest also so it will cover more information.  

Other comments are in PDF attached

Author Response

Response to Comments:

Reviewer #3

  1. The title is very interesting, its a big question to answers properly to the readers need to cite more articles related to toxicity of EO against insects

Response: Thank you for your suggestion. According to the comment, more citations on the effect of essential oils on other insect pests were added throughout the manuscript.

  1. Start with essential oil as a source of novel pesticides, i.e. EO have contact, fumigant, attractant and repellents activities against insect pests.

Response: As suggested, we have modified lines 53-54 as follows: “Essential oils (EOs) are a source of novel pesticides because they have contact, fumigant, attractant and repellents activities against several insect pests”

  1. Here are some papers you can get more information about essential oil as source of novel alternatives to chemical insecticides against various groups of insect pests.
  2. Seriphidium brevifolium essential oil: a novel alternative to synthetic insecticides against the dengue vector Aedes albopictus
  3. Fumigant toxicity and biochemical properties of (α+ β) thujone and 1, 8-cineole derived from Seriphidium brevifolium volatile oil against the red imported fire ant Solenopsis
  4. Development and evaluation of emulsifiable concentrate formulation containing Sophora alopecuroides L. extract for the novel management of Asian citrus psyllid
  5. Aniseed essential oil botanical insecticides for the management of the currant-lettuce aphid

Response: Thank you very much for your suggestion. According to the comment, we have included examples of the effect of EOs on other insect pests in the “introduction section” (Lines 54-59). In addition, we have discussed the results observed on S. frugiperda with those obtained on other species of pest insects throughout the manuscript.

  1. Need to add a Table which summarize the EO against some important pests. E.g Thymol has showed both contact, fumigant and repellent actaions.

Response: According with your suggestion, a new Table about the effect of the EOs against S. exigua, S. littoralis and S. litura, was added in the manuscript. This information is shown in SM1, and discussed in the subsection titled “Comparison of insecticidal effects of EOs between S. frugiperda and the S. littoralis-S. litura-S. exigua complex” (Lines 382-438)

Round 2

Reviewer 3 Report

The Author has improved the older version quite well. But need add more data mentioned below

Reviewer: Here are some papers you can get more information about essential oil as source of novel alternatives to chemical insecticides against various groups of insect pests. The information from below papers needs to be added in discussion and result section to support the findings. The worth of review paper is increase as you cite more related literature.

1.  Essential oils as ecofriendly biopesticides? Challenges and constraints

2. Fumigant toxicity and biochemical properties of (α+ β) thujone and 1, 8-cineole derived from Seriphidium brevifolium volatile oil against the red imported fire ant Solenopsis

  1.  Development and evaluation of emulsifiable concentrate formulation     containing Sophora alopecuroides L. extract for the novel management of       Asian citrus psyllid
  2. Aniseed essential oil botanical insecticides for the management of the currant-lettuce aphid.

Reviewer: Can the Essential Oils be a natural alternative for the control of Spodoptera frugiperda? A review of toxicity, methods, and mode of action.

Need to add a paragraph related to the Essential oils as ecofriendly biopesticides? Challenges and constraints.

Plus what are the possible formulation to increase the persistence and efficacy of EOs, as EOs  have high volatility and lost its persistence.

Author Response

Response to Comments:
Reviewer #3
1. Here are some papers you can get more information about essential oil as source of novel alternatives to chemical insecticides against various groups of insect pests. The information from below papers needs to be added in discussion and result section to support the findings. The worth of review paper is increase as you cite more related literature.
1. Essential oils as ecofriendly biopesticides? Challenges and constraints
2. Fumigant toxicity and biochemical properties of (α+ β) thujone and 1, 8-cineole derived from Seriphidium brevifolium volatile oil against the red imported fire ant Solenopsis
3. Development and evaluation of emulsifiable concentrate formulation containing Sophora alopecuroides L. extract for the novel management of Asian citrus psyllid
4. Aniseed essential oil botanical insecticides for the management of the currant-lettuce aphid.
Response: Thank you for your suggestion. According to the comment, we have included these references to the manuscript.
1. Lines 675-691
2. Lines 481-483.
3. This article was not added to the manuscript since a formulation based on methanolic plant extracts is being developed. However, similar articles, where EOs are incorporated into nanoformulations for example, have been included in section “6. Final considerations" (Lines 654-674)
4. Lines 320-323
2. Need to add a paragraph related to the Essential oils as ecofriendly biopesticides? Challenges and constraints.
Plus what are the possible formulation to increase the persistence and efficacy of EOs, as EOs have high volatility and lost its persistence.
Response: As suggested, we have included some relevant points presented in the article " Essential oils as ecofriendly biopesticides? Challenges and constraints " in the section "6. Final Considerations" as follows: “Despite EOs being classified by the US Food and Drug Administration (FDA) as Generally Recognized as Safe, we suggest that toxicity studies on other non-target organisms, animals and plants should also be performed[44]. In addition, phytotoxicity studies of EOs effects on the maize plant should accompany the mortality studies. The cost of producing EOs must also be taken into account when designing alternative strategies for pest control. Although this cost is high, there is a tendency for producers worldwide to change the cost/efficiency paradigm to one where the health of people, animals and the environment is a central issue [156,157]. Nevertheless, the number of natural pesticides in the market is still low, which may be due not only to the cost of production but also to the small number of studies carried out and with the results being applicable only in the short term. Other issues are the strict legislation hampering their incorporation into the market and the low persistence of their effects[32]. Using the knowledge of the main compounds that constitute the EOs and, in many cases, of their bioactivity, carrying out synthesizing of these compounds could lower the costs[158,159]. Furthermore, due to the high efficacy that has been demonstrated of these natural compounds, a future challenge would be for researchers and industries to work together to increase the scale of production of biopesticides and to insert them in the global market[32].” (Lines 675-691)
Moreover, we have included examples of formulations made with the aim of increasing the persistence and effectiveness of EOs: “Although these natural products have a high potential for being used as biopesticides by attacking the same sites of action as those targeted by the synthetic insecticides, studies on higher scale, such as pilot and field, should be now be carried out to determine the effectiveness of these EOs under natural conditions. In turn, due to their high volatility and photodegradation, feasible ways of applying EOs must be found to enable them to be used in cultivars[150]. Currently, several studies are being per-formed on the formulation of biodegradable nanoemulsions or encapsulates that contain EOs, or their main components, with insecticidal activity that prolongs their effects over time and facilitates their application in the field, as an alternative technology for the implementation of biopesticides in
agroecosystems. For example, the toxicity of an oil-in-water (O/W) nanoemulsion made with Tween 80 and Span 80 (non-ionic surfactants) and Cedrela odorata EOs against S. frugiperda has been determined[151]. Furthermore, it was observed that Chitosan-nanoparticles (CSNPs) loaded with S. leucantha EO decreased the activity of digestive enzymes in S. litura, H. armígera and P. xylostella[152]. Also, CSNPs loaded with Citronella EO caused the interruption of the development of S. littoralis[153]. Another innovative bioformulation, nanostructured lipid carriers (NLCs), was made using 10% w/v lipid and 10% w/v oil (L. angustifolia). This bioformulation caused high mortality and the reduced progeny of Aphis gossypii, even when applying the nanocarrier alone[154]. Thus, as can be seen, nanotechnology allows the development of bioformulations with an optimum dosage of their bioactive components in order to improve agricultural productivity, thereby generating efficient and ecofriendly alternatives[155].” (Lines 654-674)
